# BENIGN OVERFITTING IN ADVERSARIALLY ROBUST LINEAR CLASSIFICATION

## ABSTRACT

"Benign overfitting", where classifiers memorize noisy training data yet still achieve a good generalization performance, has drawn great attention in the machine learning community. To explain this surprising phenomenon, a series of works have provided theoretical justification in over-parameterized linear regression, classification, and kernel methods. However, it is not clear if benign overfitting still occurs in the presence of adversarial examples, i.e., examples with tiny and intentional perturbations to fool the classifiers. In this paper, we show that benign overfitting indeed occurs in adversarial training, a principled approach to defend against adversarial examples. In detail, we prove the risk bounds of the adversarially trained linear classifier on the mixture of sub-Gaussian data under $\ell_p$ adversarial perturbations. Our result suggests that under moderate perturbations, adversarially trained linear classifiers can achieve the near-optimal standard and adversarial risks, despite overfitting the noisy training data. Numerical experiments validate our theoretical findings.

## 1 INTRODUCTION

Modern machine learning methods such as deep learning have made many breakthroughs in a variety of application domains, including image classification (He et al., 2016; Krizhevsky et al., 2012), speech recognition (Hinton et al., 2012) and etc. These models are typically over-parameterized: the number of model parameters far exceeds the size of the training samples. One mystery is that, these over-parameterized models can memorize noisy training data and yet still achieve quite good generalization performances on the test data (Zhang et al., 2017). Many efforts have been made to explain this striking phenomenon, which against what the classical notion of overfitting might suggest. A line of research works (Soudry et al., 2018; Ji & Telgarsky, 2019b; Nacson et al., 2019; Gunasekar et al., 2018b;a) shows that there exists the so-called implicit bias (Neyshabur, 2017): the training algorithms tend to converge to certain kinds of solutions even with no explicit regularization. Specifically, Soudry et al. (2018); Ji & Telgarsky (2019b); Nacson et al. (2019) demonstrate that gradient descent trained linear classifiers on logistic or exponential loss with no regularization asymptotically converge to the maximum $L_2$ margin classifier. Recent works (Bartlett et al., 2020; Chatterji & Long, 2020; Cao et al., 2021; Wang & Thrampoulidis, 2021; Tsigler & Bartlett, 2020) further shows that over-parameterized and implicitly regularized interpolators can indeed achieve small test error, and formulate this phenomenon as "benign overfitting". More concretely, suppose the classification model $f$ is parameterized by $\boldsymbol{\theta} \in \boldsymbol{\Theta}$ and the loss is denoted as $\ell(\cdot)$. The population risk is define as

$$\mathbb{P}_{(\mathbf{x},y)\sim\mathcal{D}}[f_{\boldsymbol{\theta}}(\mathbf{x}) \neq y],$$

where data pair $(\mathbf{x}, y)$ is generated from certain data generation model. Chatterji & Long (2020) shows that with sufficient over-parameterization, gradient descent trained maximum $L_2$ margin classifier can achieve nearly optimal population risk on noisy data for data generated from a sub-Gaussian mixture model. This suggests that the overfitting can be "benign" in the over-parameterized setting.

Besides these studies on the benign overfitting phenomenon, another well-known feature of modern machine learning methods is that they are vulnerable to adversarial examples. Recent studies (Szegedy et al., 2013; Goodfellow et al., 2015) show that modern machine learning systems are brittle: slight input perturbation that is imperceptible to human eyes could mislead a well-trained classifier into wrong classification result. These malicious inputs are also known as the adversarial

examples (Szegedy et al., 2013; Goodfellow et al., 2015). Adversarial examples raise severe trustworthy issues and security concerns on the current machine learning systems especially in security-critical applications. Various methods (Kurakin et al., 2016; Madry et al., 2018; Zhang et al., 2019; Wang et al., 2019; 2020) have been proposed to defend against the threats posed by adversarial examples. One of the notable approaches is adversarial training (Madry et al., 2018). Specifically, adversarial training solves the following min-max optimization problem,

$$\min_{\boldsymbol{\theta} \in \boldsymbol{\Theta}} \frac{1}{n} \sum_{i=1}^{n} \max_{\mathbf{x}'_i \in \mathcal{B}^p_\epsilon(\mathbf{x}_i)} \ell(f_{\boldsymbol{\theta}}(\mathbf{x}'_i), y_i),$$

where $\{(\mathbf{x}_i, y_i)\}_{i=1}^{n}$ is the training set and $\mathcal{B}^p_\epsilon(\mathbf{x}_i) = \{\mathbf{x} : \|\mathbf{x} - \mathbf{x}_i\|_p \leq \epsilon\}$ denotes the $\epsilon$-ball around $\mathbf{x}_i$ in $\ell_p$ norm ($p \geq 1$). Many empirical or theoretical studies have been conducted trying to analyze or further improve adversarial training robustness (Zhang et al., 2019; Rice et al., 2020; Wang et al., 2020; Carmon et al., 2019; Wang et al., 2019; Raghunathan et al., 2020). A recent work (Sanyal et al., 2021) also pointed out that normally trained interpolators with the presence of label noise are unlikely to be adversarially robust, while adversarially robust classifiers cannot overfit noisy labels under certain conditions. However, it is still not clear whether the benign overfitting phenomenon occurs for extremely over-parameterized models in the presence of adversarial examples.

In this paper, we show that benign overfitting indeed occurs in adversarial training. In order to properly characterize the benign overfitting phenomenon on adversarial training, we also define the population adversarial risk, which is the counterpart for population risk in standard training scenario:

$$\mathbb{P}_{(\mathbf{x}, y) \sim \mathcal{D}} \left[ \exists \mathbf{x}' \in \mathcal{B}^p_\epsilon(\mathbf{x}) \ s.t., \ f_{\boldsymbol{\theta}}(\mathbf{x}') \neq y \right].$$

The adversarial risk measures the misclassification rate of the target classifier under the presence of $\ell_p$-norm adversarial perturbations. It is easy to observe that the adversarial risk is always larger than standard risk as it requires the classifier to correctly classify the data examples within the entire local $\ell_p$ norm ball.

We summarize our contributions of this paper in the following

- We show that the benign overfitting phenomenon can occur in adversarially robust linear classifiers with sufficient over-parameterization. Specifically, under moderate $\ell_p$ norm perturbations, adversarially trained linear classifiers can achieve the near-optimal standard and adversarial risks, in spite of overfitting the noisy training data.

- When the perturbation strength $\epsilon$ is set to be 0, our adversarial risk bound reduces to the standard one. The resulting standard risk bound extends Chatterji & Long (2020)'s risk bound to further characterize the behavior of the linear classifier trained by $t$-step gradient descent.

- We show that depending on the value of $p$ (perturbation norm), the adversarial risk bound can be different. The higher value of $p$ (typically for $p \geq 2$ case) actually leads to a larger gap between the adversarial risk and the standard risk with the same $\epsilon$.

**Notation.** we use lower case letters to denote scalars and lower case bold face letters to denote vectors. For a vector $\mathbf{x} \in \mathbb{R}^d$, we denote its $\ell_p$ norm ($p \geq 1$) of $\mathbf{x}$ by $\|\mathbf{x}\|_p = \left( \sum_{i=1}^{d} |x_i|^p \right)^{1/p}$, the $\ell_\infty$ norm of $\mathbf{x}$ by $\|\mathbf{x}\|_\infty = \max_{i=1}^{d} |x_i|$. We denote $\mathbf{x}^{\circ p}$ as the element-wise $p$-power of $\mathbf{x}$. For $p \geq 1$, we denote $\mathcal{B}^p_r(\mathbf{x})$ as the $\ell_p$ norm ball of radius $r$ centered at $\mathbf{x}$. Given two sequences $\{a_n\}$ and $\{b_n\}$, we write $a_n = O(b_n)$ if there exists a constant $0 < C < +\infty$ such that $a_n \leq C b_n$. We denote $a_n = \Omega(b_n)$ if $b_n = O(a_n)$. We denote $a_n = \Theta(b_n)$ if $a_n = O(b_n)$ and $a_n = \Omega(b_n)$.

## 2 RELATED WORK

There exists a large body of works on adversarial training, implicit bias and benign overfitting. In this section, we review the most relevant works with ours.

**Adversarial Training.** Adversarial training (Madry et al., 2018) and its variants (Zhang et al., 2019; Wang et al., 2019; 2020) are currently the most effective type of approaches to empirically defend against adversarial examples (Szegedy et al., 2013; Goodfellow et al., 2015). And many attempts have been made to understand its empirical success. Charles et al. (2019); Li et al. (2020) showed that the adversarially trained linear classifier directionally converges to the maximum margin classifier. Gao et al. (2019); Zhang et al. (2020b) showed that adversarial training with neural networks can achieve low robust training loss. Yet these conclusions cannot explain the test (population) performances. Another line of research focuses on the generalization performance of adversarial training

and the number of training samples. Schmidt et al. (2018) showed that adversarial models require more data than standard models to achieve certain test accuracy. Chen et al. (2020) showed that more data may actually increase the gap between the generalization error of adversarially-trained models and standard models. Yin et al. (2019); Cullina et al. (2018) studied the adversarial Rademacher complexity and VC-dimensions. Some other works focus on the trade-off between robustness and natural accuracy (Zhang et al., 2019; Tsipras et al., 2019; Wu et al., 2020; Raghunathan et al., 2020; Yang et al., 2020; Dobriban et al., 2020; Javanmard & Soltanolkotabi, 2020), adversarial model complexity lower bound (Allen-Zhu & Li, 2020), as well as the provable robustness upper bound (Fawzi et al., 2018; Zhang et al., 2020a).

Recently, some works also focus on studying the learning of robust halfspaces and linear models. Montasser et al. (2020) studied the conditions on the adversarial perturbation sets under which halfspaces are robustly learnable in the presence of random label noise. Diakonikolas et al. (2020) studied the computational complexity of adversarially robust halfspaces under $\ell_p$ norm perturbations. Zou et al. (2021) showed that adversarially trained halfspaces are provably robust with low robust classification error in the presence of noise. Dan et al. (2020) proposed an adversarial signal to noise ratio and studied the excess risk lower/upper bounds for learning Gaussian mixture models. Taheri et al. (2020); Javanmard & Soltanolkotabi (2020) studied adversarial learning of linear models on Gaussian mixture data where the data dimension and the number of training data points have a fixed ratio.

**Implicit Bias.** Several recent works studied the implicit bias of various training algorithms in over-parameterized models. Soudry et al. (2018) studied the implicit bias of gradient descent trained on linearly separable data while Ji & Telgarsky (2019b) studied the non-separable case. Gunasekar et al. (2018a) studied the implicit bias of various optimization methods in linear regression and classification problems. Ji & Telgarsky (2019a) studied the implicit bias for deep linear networks and Arora et al. (2019); Gunasekar et al. (2018b) studied the implicit bias for matrix factorization. Lyu & Li (2020) studied the implicit regularization of homogeneous neural networks with exponential loss and logistic loss.

**Benign Overfitting and Double Descent.** A series of recent works have studied the "benign over-fitting" phenomenon Bartlett et al. (2020) that when training over-parameterized models, classifiers can still achieve good population risk even when overfitting the noisy training data. Bartlett et al. (2020); Tsigler & Bartlett (2020) studied the risk bounds for over-parameterized linear (ridge) regression and showed that under certain settings, the interpolating linear model with minimum parameter norm can have asymptotically optimal risk. Chatterji & Long (2020); Cao et al. (2021); Wang & Thrampoulidis (2021) studied the risk bounds in linear logistic regression and linear support vector machines. Belkin et al. (2018; 2019a;b); Hastie et al. (2019); Wu & Xu (2020) further quantified the dependency curve between the population risk and the degree of over-parameterization and showed that the curve has a double-descent shape.

## 3 PROBLEM SETTING AND PRELIMINARIES

We consider a sub-Gaussian mixture data generation model in our work. Specifically, the clean data $(\widetilde{\mathbf{x}}, \widetilde{y}) \sim \widetilde{\mathcal{D}}$ is generated such that, for each data point $(\widetilde{\mathbf{x}}, \widetilde{y}) \in \mathbb{R}^d \times \{\pm 1\}$, we have $\widetilde{y} \sim \mathrm{Unif}(\{\pm 1\})$ and $\widetilde{\mathbf{x}} = \widetilde{y}\boldsymbol{\mu} + \boldsymbol{\xi}$ where $\boldsymbol{\xi} \in \mathbb{R}^d$ and $\xi_1, \xi_2, \ldots, \xi_d$ are i.i.d. zero-mean sub-Gaussian variables with sub-Gaussian norm at most 1. The actual data examples are sampled from a noisy distribution $\mathcal{D}$ which is close to the clean distribution $\widetilde{\mathcal{D}}$. Specifically, $\mathcal{D}$ can be any distribution over $\mathbb{R}^d \times \{\pm 1\}$ who has the same marginal distribution on $\mathbb{R}^d$ and the total variation distance $d_{\mathrm{TV}}(\mathcal{D}, \widetilde{\mathcal{D}}) \leq \eta$ where $\eta$ denotes the noise level.

Note that our data generation model is standard for studying the population risk of over-parameterized linear classification. In fact, it is exactly the same as the the one studied in Chatterji & Long (2020). In this model, following standard coupling lemma (Lindvall, 2002), there always exists a joint distribution on original data and noisy data $((\widetilde{\mathbf{x}}, \widetilde{y}), (\mathbf{x}, y))$ such that the marginal distribution for $(\widetilde{\mathbf{x}}, \widetilde{y})$ is $\widetilde{\mathcal{D}}$, the marginal distribution for $(\mathbf{x}, y)$ is $\mathcal{D}$, $\mathbb{P}[\mathbf{x} = \widetilde{\mathbf{x}}] = 1$ and $\mathbb{P}[y \neq \widetilde{y}] \leq \eta$.

In this paper, we study the problem of robust binary classification with training data $\{(\mathbf{x}_i, y_i)\}_{i=1}^n$ drawn i.i.d. from the distribution $\mathcal{D}$. Let's denote the "clean" sample index as $\mathcal{C} := \{k : y_k = \widetilde{y}_k\}$ and the "noisy" sample index as $\mathcal{N} := \{k : y_k \neq \widetilde{y}_k\}$. We consider the adversarially trained linear

---

**Algorithm 1** Gradient Descent Adversarial Training

---

1: **input:** Training data $\{\mathbf{x}_i, y_i\}_{i=1}^n$, number of training iterations $T$, maximum perturbation strength $\epsilon$, training step sizes $\alpha_t$;
2: initialize model parameter $\boldsymbol{\theta}_0 = \mathbf{0}$
3: **for** $t = 1, \ldots, T$ **do**
4:   **for** each $\{\mathbf{x}_i, y_i\}$ **do**
5:     $\mathbf{x}_i' = \operatorname{argmax}_{\mathbf{x}_i' \in \mathcal{B}_\epsilon^p(\mathbf{x}_i)} \exp(-y_i \boldsymbol{\theta}_{t-1}^\top \mathbf{x}_i')$
6:   **end for**
7:   $\boldsymbol{\theta}_t = \boldsymbol{\theta}_{t-1} - \alpha_t \cdot \nabla_{\boldsymbol{\theta}} L(\boldsymbol{\theta}_{t-1})$
8: **end for**

---

classifier under exponential loss. In such case, the adversarial loss can be explicitly written as

$$L(\boldsymbol{\theta}) = \sum_{i=1}^n \max_{\mathbf{x}_i' \in \mathcal{B}_\epsilon^p(\mathbf{x}_i)} \exp(-y_i \boldsymbol{\theta}^\top \mathbf{x}_i'). \tag{3.1}$$

In gradient descent adversarial training algorithm, the adversarial loss $L(\boldsymbol{\theta})$ is minimized by first solving the inner maximization problem in (3.1) with respect to the current model parameter $\boldsymbol{\theta}_{t-1}$ and then update the model parameter $\boldsymbol{\theta}_t$ by performing gradient descent to minimize the adversarial loss in each iteration. We summarized the training procedure for gradient descent adversarial training[1] in Algorithm 1. Note that in the linear classifier setting, the inner maximization problem in (3.1) has the following property

$$\operatorname*{argmax}_{\mathbf{x}_i' \in \mathcal{B}_\epsilon^p(\mathbf{x}_i)} \exp(-y_i \boldsymbol{\theta}^\top \mathbf{x}_i') = \operatorname*{argmax}_{\mathbf{u}_i \in \mathcal{B}_\epsilon^p(\mathbf{0})} \exp(-y_i \boldsymbol{\theta}^\top (\mathbf{x}_i + \mathbf{u}_i)) = \operatorname*{argmin}_{\|\mathbf{u}_i\|_p \le \epsilon} y_i \boldsymbol{\theta}^\top \mathbf{u}_i. \tag{3.2}$$

By Hölders' inequality it is easy to observe that the optimal adversarial loss and the corresponding gradient can be written as

$$L(\boldsymbol{\theta}) = \sum_{i=1}^n \exp(-y_i \boldsymbol{\theta}^\top \mathbf{x}_i + \epsilon \|\boldsymbol{\theta}\|_q), \nabla_{\boldsymbol{\theta}} L(\boldsymbol{\theta}) = -\sum_{i=1}^n (y_i \mathbf{x}_i - \epsilon \cdot \partial \|\boldsymbol{\theta}\|_q) \exp(-y_i \boldsymbol{\theta}^\top \mathbf{x}_i + \epsilon \|\boldsymbol{\theta}\|_q),$$

where $1/p + 1/q = 1$. Also note that in the over-parameterized settings, training examples draw from our data generation model are linearly separable with high probability (See Lemma 5.1 in Section 5). Linearly separable property ensures that the training samples have a positive margin (with high probability). Following Li et al. (2020), we also define the standard and adversarial margin as

$$\bar{\gamma} := \max_{\|\boldsymbol{\theta}\|_q = 1} \min_{i \in [n]} y_i \boldsymbol{\theta}^\top \mathbf{x}_i, \quad \gamma := \max_{\|\boldsymbol{\theta}\|_2 = 1} \min_{i \in [n]} \min_{\mathbf{x}_i' \in \mathcal{B}_\epsilon^p(\mathbf{x}_i)} y_i \boldsymbol{\theta}^\top \mathbf{x}_i', \tag{3.3}$$

which are useful in our later analysis. We also define the unique linear classifier $\theta$ that achieves adversarial margin $\gamma$ defined above as $\mathbf{w}$.

## 4 MAIN RESULTS

In this section, we study both the behavior of the population risk and the population adversarial risk for adversarially trained linear classifiers.

**Assumption 4.1.** The adversarial perturbation radius $\epsilon$ is upper bounded by a constant $R$ and is smaller than the $\ell_p$ data margin $\bar{\gamma}$, i.e., $\epsilon \le \min\{R, \bar{\gamma}\}$.

The goal of adversarial training is to obtain high-accuracy classifiers that are also robust to small input perturbations which can be ignored by human beings (e.g., small $\ell_\infty$-norm perturbations that are invisible to human eyes). Therefore, Assumption 4.1 is reasonable by constraining the maximum allowable perturbation magnitude.

**Assumption 4.2.** The noise $\boldsymbol{\xi}$ in the data generation model satisfies that $\mathbb{E}[\|\boldsymbol{\xi}\|_2^2] \ge \kappa d$ for some constant $\kappa$.

Assumption 4.2 is a common condition that has also been considered in Chatterji & Long (2020). It ensures that the summation of the variances of the data input increases in the order of $\Theta(d)$. Clearly, this assumption covers the most common setting where the entries of $\xi$ are i.i.d. and have a variance larger than or equal to $\kappa$.

---

[1]Note that in practice people often initialize $\boldsymbol{\theta}_0$ by a small random vector (e.g., Xavier initialization (Glorot & Bengio, 2010)), while we follow Li et al. (2020) and set $\boldsymbol{\theta}_0 = \mathbf{0}$ for the ease of theoretical analysis.

**Assumption 4.3.** The gradient descent starts at $\mathbf{0}$, and the step sizes are set as $\alpha_0 = 1/(Gdn)$, $\alpha_t = \alpha \leq 1/(GdnM)$ for $M = \max\{[2d + \epsilon(q-1)d^{\frac{3q-2}{2q-2}}/\gamma]\exp(-\gamma^2/(Gd) + \epsilon/G), 1\}$ and a constant $G$.

Assumption 4.3 summarizes our assumptions about the gradient descent algorithm on the adversarial loss. The learning rate conditions here are to ensure the convergence of adversarial training, and is inspired by Li et al. (2020).

We first present our theorem for standard risk of adversarial training method (Algorithm 1).

**Theorem 4.4** (Standard Risk of Adversarial Training). For any $p \in [1, +\infty)$, suppose that Assumptions 4.1, 4.2 and 4.3 hold with $\kappa \in (0, 1]$ and large enough constants $R$ and $G$. Moreover, for any $\delta \in (0, 1)$, suppose the number of training samples $n \geq C \log(1/\delta)$, the dimension $d \geq C \cdot \max\{n\|\boldsymbol{\mu}\|_2^2, n^2 \log(n/\delta)\}$, the noise level $\eta < 1/C$, and $\|\boldsymbol{\mu}\|_2^2 \geq C \max\{\log(n/\delta), \epsilon\|\boldsymbol{\mu}\|_q\}$ for a large enough constant $C$. Then with probability at least $1 - \delta$, adversarially trained linear classifier $f_{\boldsymbol{\theta}_t}$ for sufficiently large $t$ under $\ell_p$-norm $\epsilon$-perturbation satisfies the following standard risk

$$\mathbb{P}_{(\mathbf{x},y)\sim\mathcal{D}}[f_{\boldsymbol{\theta}_t}(\mathbf{x}) \neq y] \leq \eta + \exp\left(-C'\left(\frac{(\|\boldsymbol{\mu}\|_2^2 - 4\epsilon\|\boldsymbol{\mu}\|_q)}{(C''+\epsilon)\sqrt{d}} - \frac{C'''\|\boldsymbol{\mu}\|_2 \log n}{\log t}\right)^2\right),$$

where $C', C'', C''' > 0$ are absolute constants, $1/p + 1/q = 1$.

**Remark 4.5.** Theorem 4.4 presents the standard risk of adversarial training under $\ell_p$ norm perturbations. Note that adversarially trained linear classifier enjoys a bounded population risk which decreases as the number of training iterations $t$ increases. Specifically, when $t \to \infty$, we have

$$\lim_{t\to\infty} \mathbb{P}_{(\mathbf{x},y)\sim\mathcal{D}}[f_{\boldsymbol{\theta}_t}(\mathbf{x}) \neq y] \leq \eta + \exp\left(-C'\left(\frac{(\|\boldsymbol{\mu}\|_2^2 - 4\epsilon\|\boldsymbol{\mu}\|_q)}{(C''+\epsilon)\sqrt{d}}\right)^2\right). \tag{4.1}$$

**Remark 4.6.** For (4.1), consider the case when the sample size $n$ is fixed but dimension $d$ and $\|\boldsymbol{\mu}\|_2$ are growing, we discuss the conditions to reach minimum standard risk of noise level $\eta$. Note that when $1 \leq p \leq 2$ we have $q \geq 2$ and $\|\boldsymbol{\mu}\|_q \leq \|\boldsymbol{\mu}\|_2$. In this case, if $\|\boldsymbol{\mu}\|_2 = \Omega(d^{1/4})$, the standard risk will come close to the noise level $\eta$ when $d$ is sufficiently large. When $p > 2$ and therefore $q < 2$, we have $\|\boldsymbol{\mu}\|_q \leq d^{1/q-1/2}\|\boldsymbol{\mu}\|_2$. In this case, if $\|\boldsymbol{\mu}\|_2 = \Omega(d^{1/4})$ and $\epsilon = O(\|\boldsymbol{\mu}\|_2/d^{1/q-1/2})$, the standard risk will come close to the noise level $\eta$ with sufficiently large $d$. Note that our theorem condition also requires that $\|\boldsymbol{\mu}\|_2 = O(\sqrt{d})$. Therefore, in order to reach the standard risk of $\eta$, we need $\|\boldsymbol{\mu}\|_2 = \Theta(d^r)$ for some $r \in (1/4, 1/2)$.

**Remark 4.7.** Choosing $\epsilon = 0$ will reduce to the standard training case. Specifically, if we set $\epsilon = 0$ in (4.1), it reduces to the same conclusion as Theorem 3.1 in Chatterji & Long (2020). However, our result is more general, as it covers the setting of adversarial training and gives risk bounds for the linear model obtained with a finite number of gradient descent iterations.

**Theorem 4.8** (Adversarial Risk of Adversarial Training). For any $\delta \in (0, 1)$, under the same conditions as in Theorem 4.4, with probability at least $1 - \delta$, the adversarially trained linear classifier $f_{\boldsymbol{\theta}_t}$ for sufficiently large $t$ under $\ell_p$-norm $\epsilon$-perturbation satisfies the following adversarial risk if $1 \leq p \leq 2$

$$\mathbb{P}_{(\mathbf{x},y)\sim\mathcal{D}}\left[\exists \mathbf{x}' \in \mathcal{B}_\epsilon^p(\mathbf{x}) \ s.t., \ f_{\boldsymbol{\theta}}(\mathbf{x}') \neq y\right]$$
$$\leq \eta + \exp\left(-C'\left(\frac{(\|\boldsymbol{\mu}\|_2^2 - 4\epsilon\|\boldsymbol{\mu}\|_q)}{(C''+\epsilon)\sqrt{d}} - \frac{C'''\|\boldsymbol{\mu}\|_2 \log n}{\log t} - \epsilon\right)^2\right),$$

and if $p > 2$,

$$\mathbb{P}_{(\mathbf{x},y)\sim\mathcal{D}}\left[\exists \mathbf{x}' \in \mathcal{B}_\epsilon^p(\mathbf{x}) \ s.t., \ f_{\boldsymbol{\theta}}(\mathbf{x}') \neq y\right]$$
$$\leq \eta + \exp\left(-C'\left(\frac{(\|\boldsymbol{\mu}\|_2^2 - 4\epsilon\|\boldsymbol{\mu}\|_q)}{(C''+\epsilon)\sqrt{d}} - \frac{C'''\|\boldsymbol{\mu}\|_2 \log n}{\log t} - \epsilon d^{\frac{1}{q}-\frac{1}{2}}\right)^2\right),$$

where $C', C'', C''' > 0$ are absolute constants, $1/p + 1/q = 1$.

**Remark 4.9.** Theorem 4.8 shows the adversarial risk of adversarial training under $\ell_p$ norm perturbations. The major difference from the standard risk (Theorem 4.4) lies in the additional $\epsilon$ or

$\epsilon d^{1/q-1/2}$ term in the exponential function. This aligns with common sense that adversarial risk should always be larger than the standard risk. This also suggests that for larger $p$-norm ($p > 2$) perturbation, the same magnitude of perturbation would lead to a larger gap between the adversarial risk and the standard risk. In terms of the perturbation strength, we can also observe that with a larger $\epsilon$, adversarially trained classifiers obtain worse adversarial risk. This has been verified by many empirical observations of adversarial training (Madry et al., 2018; Zhang et al., 2019).

**Remark 4.10.** Note that when $t \to \infty$, if $1 \le p \le 2$, we have the following adversarial risk bound:

$$\lim_{t\to\infty} \mathbb{P}_{(\mathbf{x},y)\sim\mathcal{D}}\left[\exists \mathbf{x}' \in \mathcal{B}_\epsilon^p(\mathbf{x}), f_{\boldsymbol{\theta}}(\mathbf{x}') \neq y\right] \le \eta + \exp\left(-C'\left(\frac{(\|\boldsymbol{\mu}\|_2^2 - 4\epsilon\|\boldsymbol{\mu}\|_q)}{(C''+\epsilon)\sqrt{d}} - \epsilon\right)^2\right),$$

and if $p > 2$, we have

$$\lim_{t\to\infty} \mathbb{P}_{(\mathbf{x},y)\sim\mathcal{D}}\left[\exists \mathbf{x}' \in \mathcal{B}_\epsilon^p(\mathbf{x}), f_{\boldsymbol{\theta}}(\mathbf{x}') \neq y\right] \le \eta + \exp\left(-C'\left(\frac{(\|\boldsymbol{\mu}\|_2^2 - 4\epsilon\|\boldsymbol{\mu}\|_q)}{(C''+\epsilon)\sqrt{d}} - \epsilon d^{\frac{1}{q}-\frac{1}{2}}\right)^2\right).$$

Similar to the standard risk case (Remark 4.6), when $1 \le p \le 2$, if $\|\boldsymbol{\mu}\|_2 = \Theta(d^r)$ for some $r \in (1/4, 1/2]$, the adversarial risk will also come close to the noise level $\eta$ with sufficiently large $d$. When $p > 2$, if we have $\|\boldsymbol{\mu}\|_2 = \Theta(d^r)$ for some $r \in (1/4, 1/2]$ and $\epsilon = O(\|\boldsymbol{\mu}\|_2/d^{1/q})$, the adversarial risk will be close to $\eta$ with sufficiently large $d$. Note that compared to the standard risk, this requirement on $\epsilon$ is slightly stronger.

**Remark 4.11.** Note that our results imply a striking fact that unlike those observed in previous studies (e.g., Rice et al. (2020) showed that overfitting leads to worse empirical robustness on real image distributions), overfitting in adversarial training can be benign for certain distributions. Specifically, Remark 4.10 shows that for linear models with sub-Gaussian mixture data, the overfitting effect is indeed benign. This is later empirically verified in the experiments for both linear and neural network models.

## 5 PROOF OUTLINE OF THE MAIN RESULTS

In this section, we present the proofs of our main theorems, which consists of three main steps.

**Statistical properties of the training data points.** We first list some basic properties of the training data points based on our data model defined in Section 3.

**Lemma 5.1** (Lemma 4.7 in Chatterji & Long (2020)). Let $\mathbf{z}_k = y_k\mathbf{x}_k$. There exist absolute constants $R$, $\kappa$ and $G$ and $C$, such that if the assumptions in Theorem 4.4 hold, then with probability at least $1 - \delta$,

$$\frac{d}{c_0} \le \|\mathbf{z}_k\|_2^2 \le c_0 d \text{ for all } k \in [n], \tag{5.1}$$

$$|\mathbf{z}_i^\top \mathbf{z}_j| \le c_0\left(\|\boldsymbol{\mu}\|_2^2 + \sqrt{d\log(n/\delta)}\right) \text{ for all } i \neq j, \tag{5.2}$$

$$|\boldsymbol{\mu}^\top \mathbf{z}_k - \|\boldsymbol{\mu}\|_2^2| \le \|\boldsymbol{\mu}\|_2^2/2 \text{ for all } k \in \mathcal{C}, \tag{5.3}$$

$$|\boldsymbol{\mu}^\top \mathbf{z}_k - (-\|\boldsymbol{\mu}\|_2^2)| \le \|\boldsymbol{\mu}\|_2^2/2 \text{ for all } k \in \mathcal{N}, \tag{5.4}$$

the number of noisy samples $|\mathcal{N}| \le (\eta + c_1)n$, and all training samples are linearly separable, where $c_0 > 1$ is an absolute constant.

Lemma 5.1 directly follows Lemma 4.7 in Chatterji & Long (2020). It provides direct high probability bounds for $\|\mathbf{z}_k\|_2$ and $\boldsymbol{\mu}^\top \mathbf{z}_k$ and also suggests that $\mathbf{z}_k$ vectors are nearly pairwise orthogonal in over-parameterized settings. It also guarantees that training examples are linearly separable with high probability.

**Landscape properties of the training objective function.** Given the properties of the training data points, we proceed to establish landscape properties of the objective function $L(\boldsymbol{\theta}_1)$. The following lemma bound the loss for the adversarially trained classifier in step 1.

**Lemma 5.2.** [Theorem 3.4 in Li et al. (2020)] Under the same conditions as in Theorem 4.4, with probability at least $1 - \delta$, we have $L(\boldsymbol{\theta}_1) \le 2n$, and

$$L(\boldsymbol{\theta}_{t+1}) \le L(\boldsymbol{\theta}_t), \tag{5.5}$$

$$1 - \frac{\boldsymbol{\theta}_t^\top \mathbf{w}}{\|\boldsymbol{\theta}_t\|_2} \le \frac{c_3 \log n}{\log t} \tag{5.6}$$

for all $t > 0$, where $c_3$ is an absolute constant.

By Lemma 5.2, one can easily observe that the adversarial training loss is bounded by $2n$ along the entire training trajectory. Lemma 5.2 also suggests that when $t \to \infty$, the adversarially trained classifier $\boldsymbol{\theta}_t$ will converge in direction to the max adversarial margin classifier $\mathbf{w}$ defined in (3.3).

**Length and direction of the adversarial training iterates $\boldsymbol{\theta}_t$.** We also establish properties of the adversarial training iterates $\boldsymbol{\theta}_t$. We have the following lemmas.

**Lemma 5.3.** Under the same conditions as in Theorem 4.4, for all adversarial training iteration $t > 0$, with probability at least $1 - \delta$, we have $\|\boldsymbol{\theta}_{t+1}\|_2 \leq (\sqrt{c_0} + \epsilon)\sqrt{d} \sum_{m=0}^{t} \alpha_m L(\boldsymbol{\theta}_m)$, where $c_0$ is the absolute constant in Lemma 5.1.

Lemma 5.3 upper bound the $L_2$ norm of adversarially trained classifier $\boldsymbol{\theta}_t$ by the summation of training losses along the training trajectory.

**Lemma 5.4.** Let $\mathbf{z}_k = y_k \mathbf{x}_k$, under the same conditions as in Theorem 4.4, for all adversarial training iteration $t \geq 0$, with probability as least $1 - \delta$, we have $\max_{k=1}^{n} \exp(-\boldsymbol{\theta}_t^\top \mathbf{z}_k) \leq c_3 \min_{k=1}^{n} \exp(-\boldsymbol{\theta}_t^\top \mathbf{z}_k)$, where $c_3 > 0$ is an absolute constant.

Lemma 5.4 provides us a way to control the loss the noisy examples during the training procedure. Note that if $\max_{k=1}^{n} \exp(-\boldsymbol{\theta}_t^\top \mathbf{z}_k) \leq c_3 \min_{k=1}^{n} \exp(-\boldsymbol{\theta}_t^\top \mathbf{z}_k)$, we also have $\max_{k=1}^{n} \exp(-\boldsymbol{\theta}_t^\top \mathbf{z}_k + \epsilon\|\boldsymbol{\theta}_t\|_q) \leq c_3 \min_{k=1}^{n} \exp(-\boldsymbol{\theta}_t^\top \mathbf{z}_k + \epsilon\|\boldsymbol{\theta}_t\|_q)$. Therefore, the worst example training loss can be bounded via the best example training loss and further be bounded by the average training loss $L(\boldsymbol{\theta}_t)$. In this way, we can guarantee that those noisy examples will not have major influence on model training even in later training stages.

By using Lemmas 5.1-5.4, we establish the following key lemma for our main theorems.

**Lemma 5.5.** Under the same condition as in Theorem 4.4, with probability at least $1 - \delta$, the adversarially trained linear model parameter $\boldsymbol{\theta}_t$ satisfies

$$\frac{\boldsymbol{\mu}^\top \boldsymbol{\theta}_t}{\|\boldsymbol{\theta}_t\|_2} \geq \left(\frac{\|\boldsymbol{\mu}\|_2^2}{4} - \epsilon\|\boldsymbol{\mu}\|_q\right) \frac{1}{(\sqrt{c_0} + \epsilon)\sqrt{d}} - \frac{c_3\|\boldsymbol{\mu}\|_2 \log n}{\log t}.$$

where $c_0$ is the absolute constant in Lemma 5.1.

Lemma 5.5 provides the lower bound for the inner product of $\boldsymbol{\mu}$ and the direction of $\boldsymbol{\theta}_t$. This lemma extends Lemma 4.4 in Li et al. (2020) by considering the training iteration $t$ rather than just the converged classifier $\mathbf{w}$, and also extends to the adversarial training setting. Notice that this lower bound actually gets larger with the increase of iteration $t$.

**Finalizing the proof.** We now present the proof for Theorems 4.4 and 4.8.

*Proof of Theorem 4.4.* First, following standard coupling lemma (Lindvall, 2002), there always exists a joint distribution on original data and noisy data $((\widetilde{\mathbf{x}}, \widetilde{y}), (\mathbf{x}, y))$ such that the marginal distribution for $(\widetilde{\mathbf{x}}, \widetilde{y})$ is $\widetilde{\mathcal{D}}$, the marginal distribution for $(\mathbf{x}, y)$ is $\mathcal{D}$, $\mathbb{P}[\mathbf{x} = \widetilde{\mathbf{x}}] = 1$ and $\mathbb{P}[y \neq \widetilde{y}] \leq \eta$. Notice that the standard population risk can be written as

$$\begin{aligned}
\mathbb{P}_{(\mathbf{x},y)\sim\mathcal{D}}[f_{\boldsymbol{\theta}_t}(\mathbf{x}) \neq y] &= \mathbb{P}_{(\mathbf{x},y)\sim\mathcal{D}}[y \cdot \boldsymbol{\theta}_t^\top \mathbf{x} < 0] \\
&\leq \eta + \mathbb{P}_{(\mathbf{x},y)\sim\mathcal{D}}[y \cdot \boldsymbol{\theta}_t^\top \mathbf{x} < 0, y = \widetilde{y}] \\
&= \eta + \mathbb{P}_{(\mathbf{x},y)\sim\mathcal{D}}[\widetilde{y} \cdot \boldsymbol{\theta}_t^\top \mathbf{x} < 0],
\end{aligned} \tag{5.7}$$

where the inequality holds since $\mathbb{P}[y \neq \widetilde{y}] \leq \eta$. Since $\widetilde{y}$ is the clean label for $\mathbf{x}$, $\widetilde{y}\mathbf{x}$ follows the same distribution as $\boldsymbol{\xi} + \boldsymbol{\mu}$ and $\mathbb{E}[\widetilde{y} \cdot \boldsymbol{\theta}^\top \mathbf{x}] = \boldsymbol{\theta}^\top \boldsymbol{\mu}$. Therefore, (5.7) can be further written as

$$\begin{aligned}
\mathbb{P}_{(\mathbf{x},y)\sim\mathcal{D}}[f_{\boldsymbol{\theta}_t}(\mathbf{x}) \neq y] &\leq \eta + \mathbb{P}_{(\mathbf{x},y)\sim\mathcal{D}}\left[\widetilde{y} \cdot \boldsymbol{\theta}_t^\top \mathbf{x} - \mathbb{E}[\widetilde{y} \cdot \boldsymbol{\theta}_t^\top \mathbf{x}] < -\boldsymbol{\theta}_t^\top \boldsymbol{\mu}\right] \\
&= \eta + \mathbb{P}_{(\mathbf{x},y)\sim\mathcal{D}}\left[\boldsymbol{\theta}_t^\top (\widetilde{y}\mathbf{x} - \mathbb{E}[\widetilde{y}\mathbf{x}]) < -\boldsymbol{\theta}_t^\top \boldsymbol{\mu}\right] \\
&\leq \eta + \exp\left(-c\frac{(\boldsymbol{\theta}_t^\top \boldsymbol{\mu})^2}{\|\boldsymbol{\theta}_t\|_2^2}\right),
\end{aligned} \tag{5.8}$$

where the last inequality holds by applying a Hoeffding-type concentration inequality (Theorem C.1) with $t = (\boldsymbol{\theta}_t^\top \boldsymbol{\mu})^2$. This bound in (5.8) enables the application of Lemma 5.5 which characterizes

how the direction of $\boldsymbol{\theta}_t$ aligns with $\boldsymbol{\mu}$ during training. By direct calculation, we have

$$\mathbb{P}_{(\mathbf{x},y)\sim\mathcal{D}}[f_{\boldsymbol{\theta}_t}(\mathbf{x}) \neq y] \leq \eta + \exp\left(- c\left(\frac{\left(\frac{\|\boldsymbol{\mu}\|_2^2}{4} - \epsilon\|\boldsymbol{\mu}\|_q\right)}{(\sqrt{c_0} + \epsilon)\sqrt{d}} - \frac{c_3\|\boldsymbol{\mu}\|_2 \log n}{\log t}\right)^2\right).$$

This concludes the proof. $\qquad\square$

*Proof of Theorem 4.8.* Similar as in the proof of Theorem 4.4, we start with a calculating an upper bound of the population risk based on the formulation of the label noise. By the definition of the adversarial risk, we have

$$
\begin{aligned}
\mathbb{P}_{(\mathbf{x},y)\sim\mathcal{D}}\left[\exists \mathbf{x}' \in \mathcal{B}_\epsilon^p(\mathbf{x}) \; s.t., \; f_{\boldsymbol{\theta}_t}(\mathbf{x}') \neq y\right] &= \mathbb{P}_{(\mathbf{x},y)\sim\mathcal{D}}[\exists \mathbf{x}' \in \mathcal{B}_\epsilon^p(\mathbf{x}) \; s.t., \; y \cdot \boldsymbol{\theta}_t^\top \mathbf{x}' < 0] \\
&\leq \eta + \mathbb{P}_{(\mathbf{x},y)\sim\mathcal{D}}[\exists \mathbf{x}' \in \mathcal{B}_\epsilon^p(\mathbf{x}) \; s.t., \; y \cdot \boldsymbol{\theta}_t^\top \mathbf{x}' < 0, y = \widetilde{y}] \\
&= \eta + \mathbb{P}_{(\mathbf{x},y)\sim\mathcal{D}}\left[\min_{\mathbf{u}\in\mathcal{B}_\epsilon^p(\mathbf{0})} \widetilde{y} \cdot \boldsymbol{\theta}_t^\top (\mathbf{x} + \mathbf{u}) < 0\right] \\
&= \eta + \mathbb{P}_{(\mathbf{x},y)\sim\mathcal{D}}\left[\widetilde{y} \cdot \boldsymbol{\theta}_t^\top \mathbf{x} - \epsilon\|\boldsymbol{\theta}_t\|_q < 0\right], \qquad (5.9)
\end{aligned}
$$

where the inequality holds in the same way as in (5.7). Since $\widetilde{y}$ is the clean label for $\mathbf{x}$, $\widetilde{y}\mathbf{x}$ follows the same distribution as $\boldsymbol{\xi} + \boldsymbol{\mu}$ and $\mathbb{E}[\widetilde{y} \cdot \boldsymbol{\theta}_t^\top \mathbf{x}] = \boldsymbol{\theta}_t^\top \boldsymbol{\mu}$. Therefore, (5.9) can be further written as

$$
\begin{aligned}
\mathbb{P}_{(\mathbf{x},y)\sim\mathcal{D}}\left[\exists \mathbf{x}' \in \mathcal{B}_\epsilon^p(\mathbf{x}) \; s.t., \; f_{\boldsymbol{\theta}_t}(\mathbf{x}') \neq y\right] &\leq \eta + \mathbb{P}_{(\mathbf{x},y)\sim\mathcal{D}}\left[\widetilde{y} \cdot \boldsymbol{\theta}_t^\top \mathbf{x} - \mathbb{E}[\widetilde{y} \cdot \boldsymbol{\theta}_t^\top \mathbf{x}] < -\boldsymbol{\theta}_t^\top \boldsymbol{\mu} + \epsilon\|\boldsymbol{\theta}_t\|_q\right] \\
&= \eta + \mathbb{P}_{(\mathbf{x},y)\sim\mathcal{D}}\left[\boldsymbol{\theta}_t^\top \left(\widetilde{y}\mathbf{x} - \mathbb{E}[\widetilde{y}\mathbf{x}]\right) < -\boldsymbol{\theta}_t^\top \boldsymbol{\mu} + \epsilon\|\boldsymbol{\theta}_t\|_q\right] \\
&\leq \eta + \exp\left(- c\frac{(\boldsymbol{\theta}_t^\top \boldsymbol{\mu} - \epsilon\|\boldsymbol{\theta}_t\|_q)^2}{\|\boldsymbol{\theta}_t\|_2^2}\right), \qquad (5.10)
\end{aligned}
$$

where the second inequality holds by applying the Hoeffding-type concentration inequality (Theorem C.1) with $t = (\boldsymbol{\theta}_t^\top \boldsymbol{\mu} - \epsilon\|\boldsymbol{\theta}_t\|_q)^2$. Based on (5.10) and Lemma 5.5, we can further give bounds of the adversarial risk. We consider the two settings $1 \leq p \leq 2$ and $2 < p < \infty$ separately.

When $1 \leq p \leq 2$, we have $q \geq 2$ and $\|\boldsymbol{\theta}\|_q \leq \|\boldsymbol{\theta}\|_2$. In this case, by Lemma 5.5 we obtain

$$\mathbb{P}_{(\mathbf{x},y)\sim\mathcal{D}}[f_{\boldsymbol{\theta}_t}(\mathbf{x}) \neq y] \leq \eta + \exp\left(- c\left(\frac{\left(\frac{\|\boldsymbol{\mu}\|_2^2}{4} - \epsilon\|\boldsymbol{\mu}\|_q\right)}{(\sqrt{c_0} + \epsilon)\sqrt{d}} - \frac{c_3\|\boldsymbol{\mu}\|_2 \log n}{\log t} - \epsilon\right)^2\right).$$

When $p > 2$ and therefore $q < 2$, we have $\|\boldsymbol{\mu}\|_q \leq d^{1/q - 1/2}\|\boldsymbol{\mu}\|_2$. In this case, by Lemma 5.5 we obtain

$$\mathbb{P}_{(\mathbf{x},y)\sim\mathcal{D}}[f_{\boldsymbol{\theta}_t}(\mathbf{x}) \neq y] \leq \eta + \exp\left(- c\left(\frac{\left(\frac{\|\boldsymbol{\mu}\|_2^2}{4} - \epsilon\|\boldsymbol{\mu}\|_q\right)}{(\sqrt{c_0} + \epsilon)\sqrt{d}} - \frac{c_3\|\boldsymbol{\mu}\|_2 \log n}{\log t} - \epsilon d^{\frac{1}{q} - \frac{1}{2}}\right)^2\right).$$

This concludes the proof. $\qquad\square$

## 6 EXPERIMENTS

In this section, we experimentally study the behavior of the adversarially trained linear classifier in the over-parameterized regime on synthetic data. Specifically, we generate 50 training samples and 2000 test samples and set the label noise ratio $\eta = 0.1$ for all experiments. Each clean sample $(\widetilde{\mathbf{x}}, \widetilde{y})$ is drawn from a Gaussian mixture model such that $\widetilde{y} \sim \text{Unif}(\{\pm 1\})$ and $\widetilde{\mathbf{x}} = \widetilde{y}\boldsymbol{\mu} + \boldsymbol{\xi}$ where $\boldsymbol{\xi} \in \mathbb{R}^d$ and $\xi_1, \xi_2, \ldots, \xi_d$ are i.i.d. standard Gaussian variables and $\boldsymbol{\mu}$ simply shares the same direction as an all-one vector but has various different magnitudes. This aligns with our model assumptions in Section 3. For the adversarial training algorithm, we directly follows Algorithm 1 except using a more practical Xavier normal initialization (Glorot & Bengio, 2010), i.e., sampling $\boldsymbol{\theta}_0$ i.i.d. from from $\mathcal{N}(0, 1/\sqrt{d})$. We set the learning rate $\alpha_t = 0.001$ and the total number of iterations $T = 1000$ for all experiments. All results are obtained by averaging over 10 independent runs (both data sampling and training).

In the first set of experiments, we verify our main conclusions in this paper, that benign overfitting can occur in adversarial training. Figure 1 (a-d) illustrates the risk and the adversarial risk of adversarially trained linear classifiers versus the dimension $d$ under different scalings of $\boldsymbol{\mu}$ for both

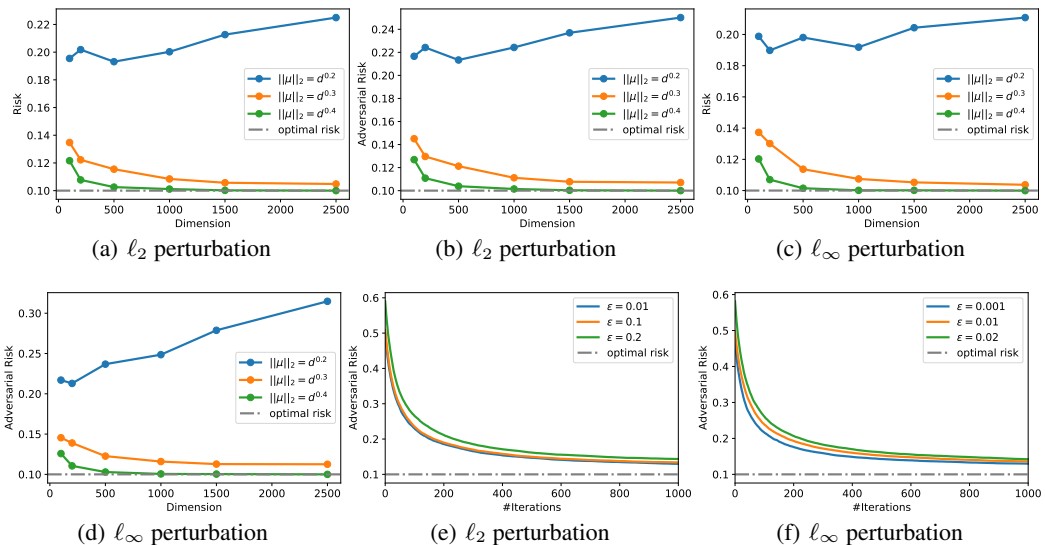

(a) $\ell_2$ perturbation      (b) $\ell_2$ perturbation      (c) $\ell_\infty$ perturbation

(d) $\ell_\infty$ perturbation      (e) $\ell_2$ perturbation      (f) $\ell_\infty$ perturbation

Figure 1: (a-d) Risk and adversarial risk of adversarially trained linear classifiers versus the dimension $d$ under different scalings of $\boldsymbol{\mu}$. (a)(b) show the results for $\ell_2$ perturbation with $\epsilon = 0.1$ and (c)(d) show the results for $\ell_\infty$ perturbation with $\epsilon = 0.01$. (e-f) Adversarial risk of adversarially trained linear classifiers versus the training iterations $t$ for different $\epsilon$ with $d = 200$ and $\|\boldsymbol{\mu}\|_2 = d^{0.3}$. The training error reaches 0 for all experiments.

$\ell_2$-norm and $\ell_\infty$-norm perturbations. We can observe that when $\|\boldsymbol{\mu}\|_2 = d^{0.2}$, the (adversarial) risk starts to increase as the dimension $d$ increases after an initial dive for both $\ell_2$-norm and $\ell_\infty$-norm perturbations. While for cases where $\|\boldsymbol{\mu}\|_2 = d^{0.3}$ and $\|\boldsymbol{\mu}\|_2 = d^{0.4}$, we can observe that the (adversarial) risk decreases steadily to the optimal risk $\eta$ as the dimension $d$ increases. This results backup our theory in Section 4 that the optimal risk is achievable when $\|\boldsymbol{\mu}\|_2 = \Theta(d^r)$ and $r \in (1/4, 1/2]$. Note that the training error reaches 0 for all settings in Figure 1.

In Figure 1 (e-f), we present the adversarial risk[2] of adversarially trained linear classifiers versus the training iterations $t$ with different $\epsilon$ but fixed dimension $d$ and $\|\boldsymbol{\mu}\|_2$ for both $\ell_2$-norm and $\ell_\infty$-norm perturbations. We can also observe that in general, a larger $\epsilon$ will lead to the worse adversarial risk of the adversarially trained classifier. This also backs up our theory in Theorem 4.8.

As our ultimate goal is to study the benign overfitting phenomenon in real-world adversarial training settings, we also conducted experiments on 2-layer neural networks with ReLU activation functions. In fact, the performances on the 2-layer ReLU network suggest very similar trends as the linear model. Due to space limit, we display these results in the supplemental materials.

## 7    CONCLUSIONS AND FUTURE WORK

In this paper, we show that the benign overfitting phenomenon also occurs in adversarial training, a principled approach to defend against adversarial examples. Specifically, we derive the risk bounds of the adversarially trained linear classifiers and show that under moderate $\ell_p$-norm perturbations, they can achieve the near-optimal standard and adversarial risks, despite overfitting the noisy training data. The numerical experimental results also validate our theoretical findings.

Our current analysis is limited to linear classifiers, while in practice, adversarial training is commonly used with neural networks. We believe our work is the first step towards analyzing benign overfitting in adversarially trained neural networks. Yet extending our current analysis to adversarially trained neural networks is highly non-trivial and we leave it as a future work.

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

## A    COMPARISON WITH DAN ET AL. (2020), TAHERI ET AL. (2020) AND JAVANMARD  SOLTANOLKOTABI (2020)

Dan et al. (2020) proposed an adversarial signal to noise ratio and studied the excess risk lower/upper bounds for learning Gaussian mixture models. Compared to the setting studied in Dan et al. (2020), our setting covers additional label flipping noises. More importantly, we study an estimator found by gradient descent that overfits the training data, while Dan et al. (2020) studied a specific plug-in estimator which does not overfit the training data. Due to these differences, there is a discrepancy in the risk bounds derived in both papers.

Taheri et al. (2020); Javanmard & Soltanolkotabi (2020) studied adversarial learning of linear models in the proportional limit setting, i.e., $d/n = O(1)$. In this setting, the data Gram matrix and the sample covariance matrix can be studied based on random matrix theory/Gaussian comparison inequalities/convex Gaussian min-max theorem. In contrast, in our setting where $d > \widetilde{O}(n^2)$, the sample covariance matrix is singular but the $n \times n$ Gram matrix concentrates around its expectation. Therefore, our setting is different from the proportional limit setting in Taheri et al. (2020); Javanmard & Soltanolkotabi (2020), and these results are not directly comparable.

## B    PROOF OF KEY TECHNICAL LEMMAS

### B.1    PROOF OF LEMMA 5.2

*Proof.* We first prove that $L(\boldsymbol{\theta}_1) \leq 2n$. To show this, we observe that $\boldsymbol{\theta}_1 = \alpha_0 \sum_{k=1}^n \mathbf{z}_k$. Therefore

$$
\begin{aligned}
L(\boldsymbol{\theta}_1) &= \sum_{k=1}^n \exp(-\boldsymbol{\theta}_1^\top \mathbf{z}_k + \epsilon \|\boldsymbol{\theta}_1\|_q) \\
&= \sum_{k=1}^n \exp\left( -\alpha_0 \sum_{i=1}^n \mathbf{z}_i^\top \mathbf{z}_k + \alpha_0 \epsilon \Big\| \sum_{i=1}^n \mathbf{z}_i \Big\|_q \right) \\
&\leq \sum_{k=1}^n \exp\left( \alpha_0 n \Big( c_0 \big( \|\boldsymbol{\mu}\|_2^2 + \sqrt{d \log(n/\delta)} \big) + \epsilon \sqrt{c_0} d \Big) \right) \\
&\leq \sum_{k=1}^n \exp(1/16) \leq 2n,
\end{aligned}
$$

where the first equality holds due to Lemma 5.1 and the fact that for any $\mathbf{u} \in \mathbb{R}^d, \|\mathbf{u}\|_q \leq \|\mathbf{u}\|_1 \leq \sqrt{d}\|\mathbf{u}\|_2$, while the second inequality is by the choice of sufficiently small $\alpha_0$ and the assumptions that $d \geq Cn\|\boldsymbol{\mu}\|_2^2$ and $\epsilon \leq R$ for some absolute constants $C$ and $R$.

The rest part of Lemma 5.2 summarizes parts of the results in Li et al. (2020). However, the results in Li et al. (2020) are derived under the setting that $\|\mathbf{x}_i\|_2 \leq 1$, Therefore to prove lemma 5.2, we re-scale our data and model parameters and convert our setting to the setting in Li et al. (2020).

By lemma 5.1, with probability at least $1 - \delta$, $\|\mathbf{x}_i\|_2^2 \leq c_0 d$ for all $i \in [n]$. We therefore denote $B := \sqrt{c_0 d}$, and then $\widetilde{\mathbf{x}}_i := \mathbf{x}_i/B$ has $\ell_2$-norm less than or equal to one. Further denote by $\boldsymbol{\beta}_t$ the linear model parameters in Li et al. (2020)'s algorithm, $\widetilde{\mathbf{z}}_i = y_i \widetilde{\mathbf{x}}_i$, $\eta_t$ as their step sizes, $\widetilde{\epsilon}$ as their perturbation strength, and

$$
\widetilde{\gamma} := \max_{\|\boldsymbol{\theta}\|_2 = 1} \min_{i \in [n]} y_i \boldsymbol{\theta}^\top \widetilde{\mathbf{x}}_i
$$

as the $\ell_p$ margin. Then the adversarial training update rule in Li et al. (2020) is

$$
\boldsymbol{\beta}_{t+1} = \boldsymbol{\beta}_t - \frac{\eta_t}{n} \sum_{i=1}^n \nabla_{\boldsymbol{\beta}} \exp(-\boldsymbol{\beta}_t^\top \widetilde{\mathbf{z}}_k + \widetilde{\epsilon}\|\boldsymbol{\beta}_t\|_q).
$$

Note that our update rule is

$$
\boldsymbol{\theta}_{t+1} = \boldsymbol{\theta}_t - \alpha_t \sum_{k=1}^n \nabla_{\boldsymbol{\theta}} \exp(-\boldsymbol{\theta}_t^\top \mathbf{z}_k + \epsilon\|\boldsymbol{\theta}_t\|_q).
$$

Now in order to apply the results in Li et al. (2020), we convert our parameters to match their scaling. Since

$$
\begin{aligned}
\boldsymbol{\theta}_{t+1} &= \boldsymbol{\theta}_t - \alpha_t \sum_i \nabla_{\boldsymbol{\theta}} \exp(-B\boldsymbol{\theta}_t^\top \mathbf{z}_k/B + \epsilon \|B\boldsymbol{\theta}_t\|_q/B) \\
&= \boldsymbol{\theta}_t - \frac{nB\alpha_t}{n} \sum_i \nabla_{(B\boldsymbol{\theta})} \exp(-B\boldsymbol{\theta}_t^\top \mathbf{z}_k/B + \epsilon \|B\boldsymbol{\theta}_t\|_q/B).
\end{aligned}
$$

Therefore

$$
B\boldsymbol{\theta}_{t+1} = B\boldsymbol{\theta}_t - \frac{nB^2\alpha_t}{n} \sum_i \nabla_{(B\boldsymbol{\theta})} \exp(-B\boldsymbol{\theta}_t^\top \mathbf{z}_k/B + \epsilon \|B\boldsymbol{\theta}_t\|_q/B).
$$

It is easy to observe that we can now apply Theorem 3.3 and Theorem 3.4 in Li et al. (2020) by setting $\boldsymbol{\beta}_t = B\boldsymbol{\theta}_t, \eta_t = nB^2\alpha_t, \widetilde{\epsilon} = \epsilon/B$. Moreover, by $\widetilde{\mathbf{x}}_i = \mathbf{x}_i/B, \widetilde{\epsilon} = \epsilon/B$ and the definition of $\widetilde{\gamma}$, we have $\widetilde{\gamma} = \bar{\gamma}/B$. Based on these relations, it is easy to see that under the conditions of Lemma 5.2, $\widetilde{\mathbf{x}}_i, \eta_t, \widetilde{\epsilon}, \widetilde{\gamma}$ satisfy the assumptions of Theorems 3.3 and 3.4 in Li et al. (2020). Now (5.5) is an intermediate result of the proof of Theorem 3.3 in Li et al. (2020), and (5.6) follows by Theorem 3.4 in Li et al. (2020). $\qquad\square$

## B.2 PROOF OF LEMMA 5.3

*Proof.* We have

$$
\begin{aligned}
\|\boldsymbol{\theta}_{t+1}\|_2 &= \left\| \sum_{m=0}^t \alpha_m \cdot \nabla L(\boldsymbol{\theta}_m) \right\|_2 \\
&\leq \sum_{m=0}^t \alpha_m \|\nabla L(\boldsymbol{\theta}_m)\|_2 \\
&\leq \sum_{m=0}^t \alpha_m \left\| \sum_{k=1}^n (\mathbf{z}_k - \epsilon \cdot \partial\|\boldsymbol{\theta}_m\|_q) \cdot \exp\left(-\mathbf{z}_k^\top \boldsymbol{\theta}_m + \epsilon\|\boldsymbol{\theta}_m\|_q\right) \right\|_2,
\end{aligned}
$$

where the first three inequality hold by triangle inequality. By Lemma C.2, we have

$$
\begin{aligned}
\|\boldsymbol{\theta}_{t+1}\|_2 &\leq \sum_{m=0}^t \alpha_m \sum_{k=1}^n (\|\mathbf{z}_k\|_2 + \epsilon\sqrt{d}) \cdot \exp\left(-\mathbf{z}_k^\top \boldsymbol{\theta}_m + \epsilon\|\boldsymbol{\theta}_m\|_q\right) \\
&\leq (\sqrt{c_0} + \epsilon)\sqrt{d} \sum_{m=0}^t \alpha_m \sum_{k=1}^n \cdot \exp\left(-\mathbf{z}_k^\top \boldsymbol{\theta}_m + \epsilon\|\boldsymbol{\theta}_m\|_q\right) \\
&= (\sqrt{c_0} + \epsilon)\sqrt{d} \sum_{m=0}^t \alpha_m L(\boldsymbol{\theta}_m),
\end{aligned}
$$

where the second inequality is due to Lemma 5.1. $\qquad\square$

## B.3 PROOF OF LEMMA 5.4

*Proof.* We will prove this lemma by induction.

Let's denote $E_k^t = \exp(-\boldsymbol{\theta}_t^\top \mathbf{z}_k)$. Without loss of generality, let $E_1^t$ denotes the maximum of $\{E_k^t\}_{k=1}^n$ and $E_2^t$ denotes the minimum of $\{E_k^t\}_{k=1}^n$. We also define $A_t := E_1^t/E_2^t$ and the goal is to show that $A_t \leq 5c_0^2$.

For the base case ($t = 0$), we have $E_k^0 = \exp(0) = 1$. Therefore we have $A_0 = 1 \leq 5c_0^2$.

For $t > 0$, notice that

$$
\begin{aligned}
A_{t+1} &= \frac{\exp(-\boldsymbol{\theta}_{t+1}^\top \mathbf{z}_1)}{\exp(-\boldsymbol{\theta}_{t+1}^\top \mathbf{z}_2)} = \frac{\exp(-\boldsymbol{\theta}_t^\top \mathbf{z}_1)}{\exp(-\boldsymbol{\theta}_t^\top \mathbf{z}_2)} \cdot \frac{\exp(\alpha_t \nabla L(\boldsymbol{\theta}_t)^\top \mathbf{z}_1)}{\exp(\alpha_t \nabla L(\boldsymbol{\theta}_t)^\top \mathbf{z}_2)} \\
&= A_t \cdot \frac{\exp(-\alpha_t \sum_{k=1}^n (\mathbf{z}_k - \epsilon \partial \|\boldsymbol{\theta}_t\|_q)^\top \mathbf{z}_1 \cdot \exp(-\boldsymbol{\theta}_t^\top \mathbf{z}_k + \epsilon\|\boldsymbol{\theta}_t\|_q))}{\exp(-\alpha_t \sum_{k=1}^n (\mathbf{z}_k - \epsilon \partial \|\boldsymbol{\theta}_t\|_q)^\top \mathbf{z}_2 \cdot \exp(-\boldsymbol{\theta}_t^\top \mathbf{z}_k + \epsilon\|\boldsymbol{\theta}_t\|_q))} \\
&= A_t \cdot \underbrace{\frac{\exp(-\alpha_t (\mathbf{z}_1 - \epsilon \partial \|\boldsymbol{\theta}_t\|_q)^\top \mathbf{z}_1 \cdot \exp(-\boldsymbol{\theta}_t^\top \mathbf{z}_k + \epsilon\|\boldsymbol{\theta}_t\|_q))}{\exp(-\alpha_t (\mathbf{z}_2 - \epsilon \partial \|\boldsymbol{\theta}_t\|_q)^\top \mathbf{z}_2 \cdot \exp(-\boldsymbol{\theta}_t^\top \mathbf{z}_k + \epsilon\|\boldsymbol{\theta}_t\|_q))}}_{I_1} \\
&\quad \cdot \underbrace{\frac{\exp(-\alpha_t \sum_{k\neq 1}^n (\mathbf{z}_k - \epsilon \partial \|\boldsymbol{\theta}_t\|_q)^\top \mathbf{z}_1 \cdot \exp(-\boldsymbol{\theta}_t^\top \mathbf{z}_k + \epsilon\|\boldsymbol{\theta}_t\|_q))}{\exp(-\alpha_t \sum_{k\neq 2}^n (\mathbf{z}_k - \epsilon \partial \|\boldsymbol{\theta}_t\|_q)^\top \mathbf{z}_2 \cdot \exp(-\boldsymbol{\theta}_t^\top \mathbf{z}_k + \epsilon\|\boldsymbol{\theta}_t\|_q))}}_{I_2} \cdot
\end{aligned}
\tag{B.1}
$$

For term $I_1$, note that by Lemma 5.1 we have

$$
\sqrt{\frac{d}{c_0}} \leq \|\mathbf{z}_k\|_2 \leq \sqrt{c_0 d}.
$$

Also since by Lemma C.2, we have $\big\|\partial \|\boldsymbol{\theta}_t\|_q \big\|_p = 1$,

$$
|\mathbf{z}_k^\top \partial \|\boldsymbol{\theta}_t\|_q| \leq \|\mathbf{z}_k\|_q \cdot \big\|\partial \|\boldsymbol{\theta}_t\|_q \big\|_p = \|\mathbf{z}_k\|_q \leq \|\mathbf{z}_k\|_1 \leq \sqrt{d}\|\mathbf{z}_k\|_2 \leq \sqrt{c_0 d}.
\tag{B.2}
$$

Therefore, we have

$$
\begin{aligned}
I_1 &\leq \exp\left(-\alpha_t \left(\frac{d}{c_0} - \epsilon\sqrt{c_0}d\right) \exp(-\boldsymbol{\theta}_t^\top \mathbf{z}_1 + \epsilon\|\boldsymbol{\theta}_t\|_q) + \alpha_t \left(c_0 d + \epsilon\sqrt{c_0}d\right) \exp(-\boldsymbol{\theta}_t^\top \mathbf{z}_2 + \epsilon\|\boldsymbol{\theta}_t\|_q)\right) \\
&= \exp\left(-\alpha_t E_2^t \left(\left(\frac{d}{c_0} - \epsilon\sqrt{c_0}d\right) A_t - \left(c_0 d + \epsilon\sqrt{c_0}d\right)\right) \exp\left(\epsilon\|\boldsymbol{\theta}_t\|_q\right)\right).
\end{aligned}
\tag{B.3}
$$

For term $I_2$, by (5.2) and (B.2) we have

$$
\begin{aligned}
I_2 &\leq \exp\left(\alpha_t \left(c_0 \big(\|\boldsymbol{\mu}\|_2^2 + \sqrt{d\log(n/\delta)}\big) + \epsilon\sqrt{c_0}d\right) \left(\sum_{k\neq 1}^n \exp(-\boldsymbol{\theta}_t^\top \mathbf{z}_k + \epsilon\|\boldsymbol{\theta}_t\|_q) + \sum_{k\neq 2}^n \exp(-\boldsymbol{\theta}_t^\top \mathbf{z}_k + \epsilon\|\boldsymbol{\theta}_t\|_q)\right)\right) \\
&\leq \exp\left(2\alpha_t L(\boldsymbol{\theta}_t)\left(c_0 \big(\|\boldsymbol{\mu}\|_2^2 + \sqrt{d\log(n/\delta)}\big) + \epsilon\sqrt{c_0}d\right)\right)
\end{aligned}
\tag{B.4}
$$

Substitute (B.3) and (B.4) into (B.1), we have

$$
\begin{aligned}
A_{t+1} &\leq A_t \cdot \exp\left(-\alpha_t E_2^t \left(\left(\frac{d}{c_0} - \epsilon\sqrt{c_0}d\right) A_t - \left(c_0 d + \epsilon\sqrt{c_0}d\right)\right) \exp\left(\epsilon\|\boldsymbol{\theta}_t\|_q\right)\right) \\
&\quad \cdot \exp\left(2\alpha_t L(\boldsymbol{\theta}_t)\left(c_0 \big(\|\boldsymbol{\mu}\|_2^2 + \sqrt{d\log(n/\delta)}\big) + \epsilon\sqrt{c_0}d\right)\right).
\end{aligned}
\tag{B.5}
$$

Let us consider two cases here. If $(d/c_0 - \epsilon\sqrt{c_0}d)A_t - (c_0 d + \epsilon\sqrt{c_0}d) > c_0 d$, i.e., $A_t > (2c_0 + \epsilon\sqrt{c_0})/(1/c_0 - \epsilon\sqrt{c_0})$, we further have

$$
\begin{aligned}
A_{t+1} &\leq A_t \cdot \exp\left(-\alpha_t E_2^t c_0 d \exp\left(\epsilon\|\boldsymbol{\theta}_t\|_q\right)\right) \cdot \exp\left(2\alpha_t L(\boldsymbol{\theta}_t)\left(c_0 \big(\|\boldsymbol{\mu}\|_2^2 + \sqrt{d\log(n/\delta)}\big) + \epsilon\sqrt{c_0}d\right)\right) \\
&\leq A_t \cdot \exp\left(-\alpha_t E_2^t c_0 d \exp\left(\epsilon\|\boldsymbol{\theta}_t\|_q\right)\right) \\
&\quad \cdot \exp\left(2\alpha_t n E_2^t \left(c_0 \big(\|\boldsymbol{\mu}\|_2^2 + \sqrt{d\log(n/\delta)}\big) + \epsilon\sqrt{c_0}d\right) \exp\left(\epsilon\|\boldsymbol{\theta}_t\|_q\right)\right) \\
&= A_t \cdot \exp\left(-\alpha_t E_2^t c_0 \big(d - 2n\|\boldsymbol{\mu}\|_2^2 - 2n\sqrt{d\log(n/\delta)} - 2n\epsilon\sqrt{c_0}\big) \exp\left(\epsilon\|\boldsymbol{\theta}_t\|_q\right)\right) \\
&\leq A_t,
\end{aligned}
$$

where the second inequality is due to the fact that $L(\boldsymbol{\theta}_t) = \sum_{k=1}^{n} E_k^t \exp\left(\epsilon\|\boldsymbol{\theta}_t\|_q\right)$ and $E_2^t = \max_k E_k^t$ while the last inequality holds since $d \geq C \cdot \max\{n\|\boldsymbol{\mu}\|_2^2, n^2 \log(n/\delta)\}$.

On the other hand, if $A_t \leq (2c_0 + \epsilon\sqrt{c_0})/(1/c_0 - \epsilon\sqrt{c_0})$, we have

$$
\begin{aligned}
A_{t+1} &\leq A_t \cdot \exp\left(\alpha_t E_2^t \left(c_0 d + \epsilon\sqrt{c_0}d\right)\exp\left(\epsilon\|\boldsymbol{\theta}_t\|_q\right)\right)\\
&\quad \cdot \exp\left(2\alpha_t L(\boldsymbol{\theta}_t)\left(c_0\left(\|\boldsymbol{\mu}\|_2^2 + \sqrt{d\log(n/\delta)}\right) + \epsilon\sqrt{c_0}d\right)\right)\\
&\leq A_t \cdot \exp\left(\alpha_t L(\boldsymbol{\theta}_t)\left(c_0 d + \epsilon\sqrt{c_0}d\right)\right) \cdot \exp\left(2\alpha_t L(\boldsymbol{\theta}_t)\left(c_0\left(\|\boldsymbol{\mu}\|_2^2 + \sqrt{d\log(n/\delta)}\right) + \epsilon\sqrt{c_0}d\right)\right)\\
&\leq A_t \cdot \exp\left(2\alpha_t n\left(c_0\left(2\|\boldsymbol{\mu}\|_2^2 + 2\sqrt{d\log(n/\delta)} + d\right) + 3\epsilon\sqrt{c_0}d\right)\right)\\
&\leq (2c_0 + \epsilon\sqrt{c_0})/(1/c_0 - \epsilon\sqrt{c_0}) \cdot \exp(1/8)\\
&\leq 5c_0^2,
\end{aligned}
$$

where the first inequality is due to the fact that $A_t > 0$, the third inequality holds by Lemma 5.2, the fourth inequality is because $\alpha_t \leq 1/(c_0 C n d)$ and $d \geq C \cdot \max\{n\|\boldsymbol{\mu}\|_2^2, n^2 \log(n/\delta)\}$ and the last inequality is because $\epsilon < C'$ and $C'$ can be chosen such that $C' \leq 1/(2c_0^{1.5})$ and we have $1/c_0 - \epsilon\sqrt{c_0} > 1/(2c_0)$.

This concludes the proof. $\qquad\square$

### B.4 PROOF OF LEMMA 5.5

*Proof.* Note that

$$
\begin{aligned}
\boldsymbol{\mu}^\top \boldsymbol{\theta}_{t+1} &= \boldsymbol{\mu}^\top\left(\boldsymbol{\theta}_t + \alpha_t \sum_{k=1}^{n}\left(\mathbf{z}_k - \epsilon\partial\|\boldsymbol{\theta}_t\|_q\right)\exp(-\boldsymbol{\theta}_t^\top\mathbf{z}_k + \epsilon\|\boldsymbol{\theta}\|_1)\right)\\
&= \boldsymbol{\mu}^\top\boldsymbol{\theta}_t - \alpha_t\epsilon \cdot \boldsymbol{\mu}^\top\partial\|\boldsymbol{\theta}_t\|_q \cdot L(\boldsymbol{\theta}_t) + \alpha_t \sum_{k=1}^{n}\left(\boldsymbol{\mu}^\top\mathbf{z}_k\right)\exp(-\boldsymbol{\theta}_t^\top\mathbf{z}_k + \epsilon\|\boldsymbol{\theta}\|_q)\\
&\geq \boldsymbol{\mu}^\top\boldsymbol{\theta}_t - \alpha_t\epsilon\|\boldsymbol{\mu}\|_q \cdot L(\boldsymbol{\theta}_t) + \alpha_t \sum_{k\in\mathcal{C}}\left(\boldsymbol{\mu}^\top\mathbf{z}_k\right)\exp(-\boldsymbol{\theta}_t^\top\mathbf{z}_k + \epsilon\|\boldsymbol{\theta}\|_q)\\
&\quad + \alpha_t \sum_{k\in\mathcal{N}}\left(\boldsymbol{\mu}^\top\mathbf{z}_k\right)\exp(-\boldsymbol{\theta}_t^\top\mathbf{z}_k + \epsilon\|\boldsymbol{\theta}\|_q),
\end{aligned}
\tag{B.6}
$$

where the inequality holds in the same way as in (B.2). By Lemma 5.1 ((5.3) and (5.4)), we further bound (B.6) by

$$
\begin{aligned}
\boldsymbol{\mu}^\top\boldsymbol{\theta}_{t+1} &\geq \boldsymbol{\mu}^\top\boldsymbol{\theta}_t - \alpha_t\epsilon\|\boldsymbol{\mu}\|_q \cdot L(\boldsymbol{\theta}_t) + \frac{\alpha_t}{2}\sum_{k\in\mathcal{C}}\|\boldsymbol{\mu}\|_2^2 \exp(-\boldsymbol{\theta}_t^\top\mathbf{z}_k + \epsilon\|\boldsymbol{\theta}\|_q)\\
&\quad - \frac{3\alpha_t}{2}\sum_{k\in\mathcal{N}}\|\boldsymbol{\mu}\|_2^2 \exp(-\boldsymbol{\theta}_t^\top\mathbf{z}_k + \epsilon\|\boldsymbol{\theta}\|_q)\\
&= \boldsymbol{\mu}^\top\boldsymbol{\theta}_t - \alpha_t\epsilon\|\boldsymbol{\mu}\|_q \cdot L(\boldsymbol{\theta}_t) + \frac{\alpha_t}{2}\|\boldsymbol{\mu}\|_2^2 L(\boldsymbol{\theta}_t) - 2\alpha_t\|\boldsymbol{\mu}\|_2^2 \sum_{k\in\mathcal{N}}\exp(-\boldsymbol{\theta}_t^\top\mathbf{z}_k + \epsilon\|\boldsymbol{\theta}\|_q).
\end{aligned}
\tag{B.7}
$$

Note that we have

$$
\begin{aligned}
\sum_{k\in\mathcal{N}}\exp(-\boldsymbol{\theta}_t^\top\mathbf{z}_k + \epsilon\|\boldsymbol{\theta}\|_q) &= \sum_{k\in\mathcal{N}}\exp(-\boldsymbol{\theta}_t^\top\mathbf{z}_k)\cdot\exp(\epsilon\|\boldsymbol{\theta}\|_q)\\
&\leq c_3(\eta + c_1)n \cdot \left(\max_k E_k\right)\cdot\exp(\epsilon\|\boldsymbol{\theta}\|_q)\\
&\leq c_3(\eta + c_1)L(\boldsymbol{\theta}_t)\\
&\leq \frac{1}{8}L(\boldsymbol{\theta}_t),
\end{aligned}
$$

where the first inequality is due to Lemma 5.2 and the last inequality is because $\eta < 1/C$ and $c_1$ can be chosen arbitrarily small given sufficient large $C$. Therefore, (B.7) can be further written as

$$
\boldsymbol{\mu}^\top \boldsymbol{\theta}_{t+1} \geq \boldsymbol{\mu}^\top \boldsymbol{\theta}_t - \alpha_t \epsilon \|\boldsymbol{\mu}\|_q \cdot L(\boldsymbol{\theta}_t) + \frac{\alpha_t}{2} \|\boldsymbol{\mu}\|_2^2 L(\boldsymbol{\theta}_t) - \frac{\alpha_t}{4} \|\boldsymbol{\mu}\|_2^2 L(\boldsymbol{\theta}_t)
$$

$$
= \boldsymbol{\mu}^\top \boldsymbol{\theta}_t + \alpha_t \left( \frac{\|\boldsymbol{\mu}\|_2^2}{4} - \epsilon \|\boldsymbol{\mu}\|_q \right) \cdot L(\boldsymbol{\theta}_t)
$$

$$
= \left( \frac{\|\boldsymbol{\mu}\|_2^2}{4} - \epsilon \|\boldsymbol{\mu}\|_q \right) \cdot \sum_{m=0}^{t} \alpha_m L(\boldsymbol{\theta}_m), \tag{B.8}
$$

where the last equality is due the fact that $\boldsymbol{\theta}_0 = \mathbf{0}$. Now we multiply $\|\mathbf{w}\|_2 / \|\boldsymbol{\theta}_{t+1}\|_2$ on both sides of (B.8) and take $t \to \infty$

$$
\lim_{t \to \infty} \frac{\|\mathbf{w}\|_2 (\boldsymbol{\mu}^\top \boldsymbol{\theta}_{t+1})}{\|\boldsymbol{\theta}_{t+1}\|_2} \geq \lim_{t \to \infty} \left( \frac{\|\boldsymbol{\mu}\|_2^2}{4} - \epsilon \|\boldsymbol{\mu}\|_q \right) \frac{\|\mathbf{w}\|_2}{\|\boldsymbol{\theta}_{t+1}\|_2} \cdot \sum_{m=0}^{t} \alpha_m L(\boldsymbol{\theta}_m).
$$

Since $\|\mathbf{w}\|_2 = 1$, and by Lemma 5.2, it is easy to observe that $\mathbf{w} = \lim_{t \to \infty} \boldsymbol{\theta}_t / \|\boldsymbol{\theta}_t\|_2$, we have

$$
\boldsymbol{\mu}^\top \mathbf{w} \geq \left( \frac{\|\boldsymbol{\mu}\|_2^2}{4} - \epsilon \|\boldsymbol{\mu}\|_q \right) \cdot \lim_{t \to \infty} \frac{\sum_{m=0}^{t} \alpha_m L(\boldsymbol{\theta}_m)}{\|\boldsymbol{\theta}_{t+1}\|_2}
$$

$$
\geq \left( \frac{\|\boldsymbol{\mu}\|_2^2}{4} - \epsilon \|\boldsymbol{\mu}\|_q \right) \frac{1}{(\sqrt{c_0} + \epsilon)\sqrt{d}}.
$$

where the last inequality is due to Lemma 5.3. Note that Lemma 5.2 also suggests that $\|\boldsymbol{\theta}_t / \|\boldsymbol{\theta}_t\|_2 - \mathbf{w}\|_2 \leq c_3 \log n / \log t$, we have

$$
\boldsymbol{\mu}^\top \mathbf{w} = \boldsymbol{\mu}^\top \left( \mathbf{w} - \frac{\boldsymbol{\theta}_t}{\|\boldsymbol{\theta}_t\|_2} + \frac{\boldsymbol{\theta}_t}{\|\boldsymbol{\theta}_t\|_2} \right)
$$

$$
\leq \|\boldsymbol{\mu}\|_2 \cdot \left\| \mathbf{w} - \frac{\boldsymbol{\theta}_t}{\|\boldsymbol{\theta}_t\|_2} \right\|_2 + \frac{\boldsymbol{\mu}^\top \boldsymbol{\theta}_t}{\|\boldsymbol{\theta}_t\|_2}
$$

$$
\leq \frac{c_3 \|\boldsymbol{\mu}\|_2 \log n}{\log t} + \frac{\boldsymbol{\mu}^\top \boldsymbol{\theta}_t}{\|\boldsymbol{\theta}_t\|_2}.
$$

Therefore,

$$
\frac{\boldsymbol{\mu}^\top \boldsymbol{\theta}_t}{\|\boldsymbol{\theta}_t\|_2} \geq \boldsymbol{\mu}^\top \mathbf{w} - \frac{c_3 \|\boldsymbol{\mu}\|_2 \log n}{\log t} \geq \left( \frac{\|\boldsymbol{\mu}\|_2^2}{4} - \epsilon \|\boldsymbol{\mu}\|_q \right) \frac{1}{(\sqrt{c_0} + \epsilon)\sqrt{d}} - \frac{c_3 \|\boldsymbol{\mu}\|_2 \log n}{\log t}.
$$

$\square$

## C  AUXILIARY LEMMAS

**Theorem C.1** (Proposition 5.10 in Vershynin (2010)). Let $X_1, X_2, \ldots, X_n$ be independent centered sub-Gaussian random variables, and let $K = \max_i \|X_i\|_{\psi_2}$. Then for every $a = (a_1, a_2, \ldots, a_n) \in \mathbb{R}^n$ and for every $t > 0$, we have

$$
\mathbb{P}\left( \left| \sum_{i=1}^{n} a_i X_i \right| > t \right) \leq \exp\left( -\frac{Ct^2}{K^2 \|a\|_2^2} \right),
$$

where $C > 0$ is a constant.

**Lemma C.2.** For any $\boldsymbol{\theta} \in \mathbb{R}^d$,

$$
\left\| \partial \|\boldsymbol{\theta}\|_q \right\|_2 \leq \sqrt{d}, \ \left\| \partial \|\boldsymbol{\theta}\|_q \right\|_p = 1.
$$

*Proof.* Note that we have

$$
(\partial \|\boldsymbol{\theta}\|_q)_i = \frac{\theta_i^{q-1}}{\|\boldsymbol{\theta}\|_q^{q-1}} \cdot \text{sign}(\boldsymbol{\theta}),
$$

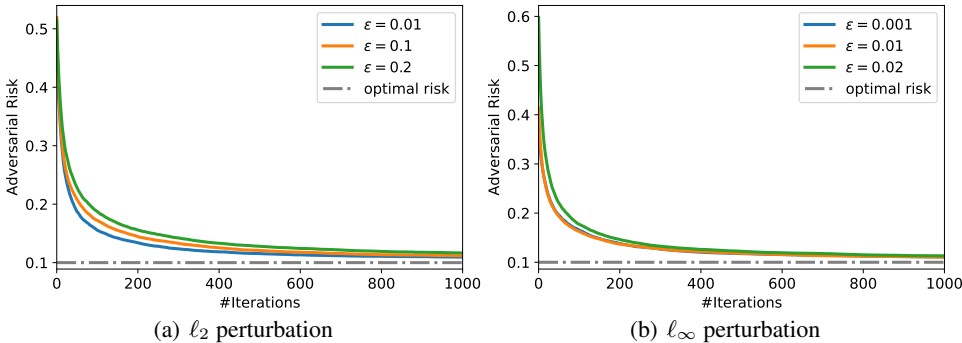

(a) $\ell_2$ perturbation  (b) $\ell_\infty$ perturbation

Figure 2: Risk and adversarial risk of adversarially trained linear classifiers versus the training iterations $t$ for different perturbation level $\epsilon$. The label noise level is set as $\eta = 0.1$, the training set size $n = 50$, dimension $d = 200$ and $\|\boldsymbol{\mu}\|_2 = d^{0.4}$. The train error reaches 0 for all experiments.

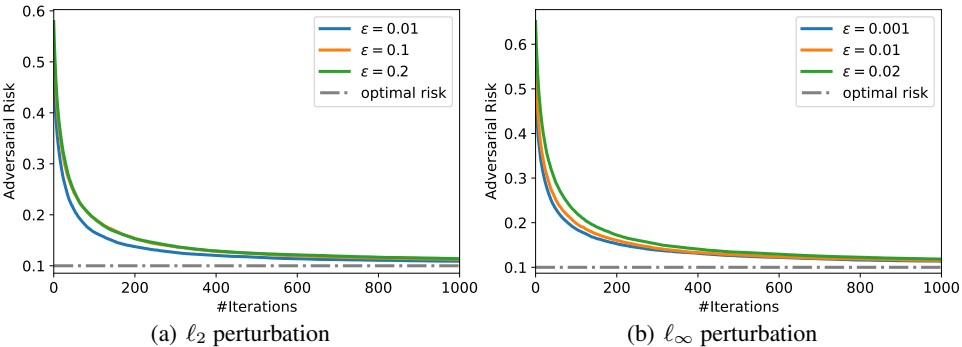

(a) $\ell_2$ perturbation  (b) $\ell_\infty$ perturbation

Figure 3: Risk and adversarial risk of adversarially trained linear classifiers versus the training iterations $t$ for different perturbation level $\epsilon$. The label noise level is set as $\eta = 0.1$, the training set size $n = 50$, dimension $d = 1000$ and $\|\boldsymbol{\mu}\|_2 = d^{0.3}$. The train error reaches 0 for all experiments.

and since for any vector $\mathbf{u} \in \mathbb{R}^d$, $\|\mathbf{u}\|_q \geq \|\mathbf{u}\|_\infty$, $\|\mathbf{u}\|_2 \leq \sqrt{d}\|\mathbf{u}\|_\infty$, we have

$$\big\|\partial\|\boldsymbol{\theta}\|_q\big\|_2 = \frac{\big\|\boldsymbol{\theta}^{\circ(q-1)}\big\|_2}{\|\boldsymbol{\theta}\|_q^{q-1}} \leq \frac{\sqrt{d}\|\boldsymbol{\theta}\|_\infty^{q-1}}{\|\boldsymbol{\theta}\|_q^{q-1}} \leq \sqrt{d},$$

where $\circ$ denotes element-wise power. This concludes the first part of the lemma. For the second part, by $p$-norm definition we have

$$\big\|\partial\|\boldsymbol{\theta}\|_q\big\|_p = \frac{\big\|\boldsymbol{\theta}^{\circ(q-1)}\big\|_p}{\|\boldsymbol{\theta}\|_q^{q-1}} = \frac{1}{\|\boldsymbol{\theta}\|_q^{q-1}}\Big(\sum_{i=1}^d (\theta_i^{q-1})^p\Big)^{1/p} = \frac{1}{\|\boldsymbol{\theta}\|_q^{q-1}}\bigg(\Big(\sum_{i=1}^d \theta_i^q\Big)^{1/q}\bigg)^{q-1} = 1.$$

$\square$

# D  ADDITIONAL EXPERIMENTS

In this section, we present the additional experiments covering more settings as well as more complex models such as 2-layer neural network.

## D.1  ADVERSARIALLY TRAINED LINEAR CLASSIFIER UNDER VARIOUS SETTINGS

In Figures 2,3,4, we plot the adversarial risk of adversarially trained linear classifiers versus the training iterations $t$ for different perturbation level $\epsilon$ for various combinations of dimension $d$ and $\|\boldsymbol{\mu}\|_2$. Specifically, in Figure 4, we can observe that with moderate perturbations and sufficient over-parameterization, adversarially trained linear classifiers can achieve near-optimal adversarial risks.

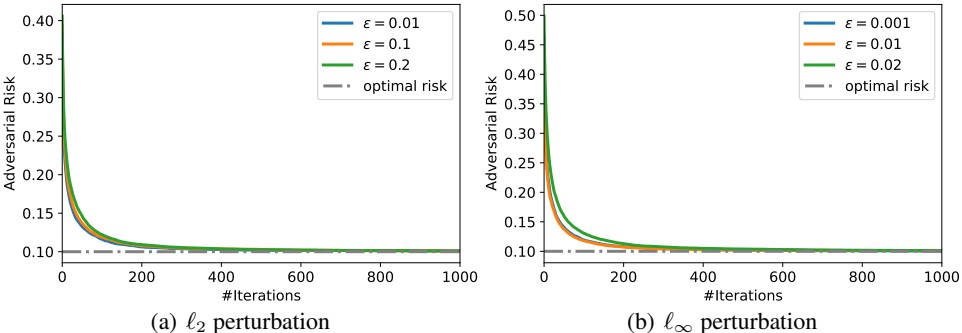

(a) $\ell_2$ perturbation

(b) $\ell_\infty$ perturbation

Figure 4: Risk and adversarial risk of adversarially trained linear classifiers versus the training iterations $t$ for different perturbation level $\epsilon$. The label noise level is set as $\eta = 0.1$, the training set size $n = 50$, dimension $d = 1000$ and $\|\boldsymbol{\mu}\|_2 = d^{0.4}$. The train error reaches $0$ for all experiments.

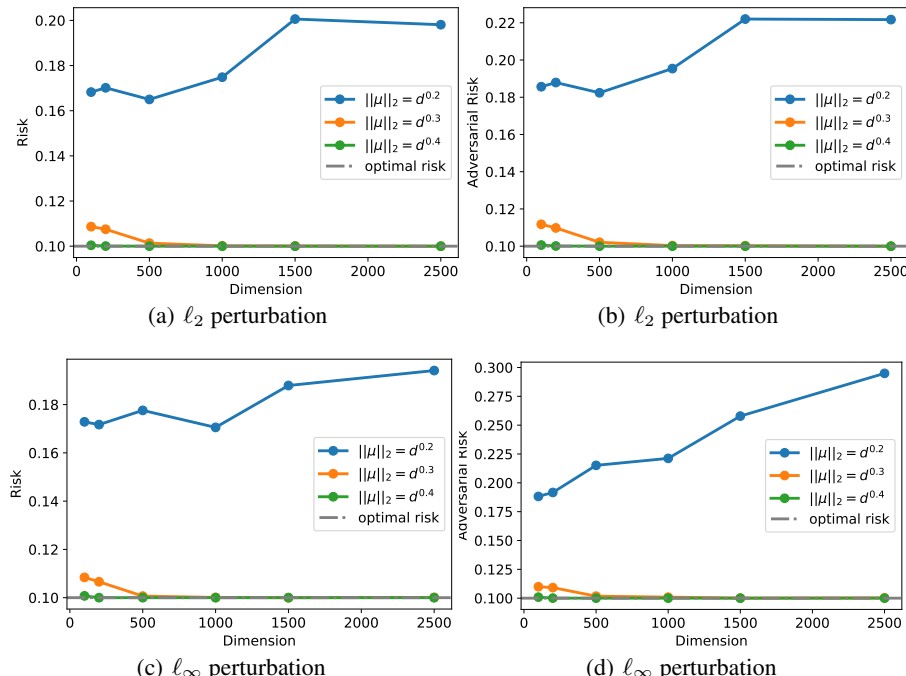

(a) $\ell_2$ perturbation

(b) $\ell_2$ perturbation

(c) $\ell_\infty$ perturbation

(d) $\ell_\infty$ perturbation

Figure 5: Risk and adversarial risk of adversarially trained 2-layer ReLU network versus the dimension $d$ under different scalings of $\boldsymbol{\mu}$. (a)(b) show the results for $\ell_2$ perturbation with $\epsilon = 0.1$ and (c)(d) show the results for $\ell_\infty$ perturbation with $\epsilon = 0.01$. The training error reaches $0$ for all experiments.

## D.2 ADVERSARIALLY TRAINED 2-LAYER NEURAL NETWORKS

We have also conducted extra experiments on 2-layer neural networks with ReLU activation functions (one extra fix-dimension hidden layer). The data generation process are the same as our linear experiments. Note that in this setting, we no longer have the closed-form solutions to the inner maximization problem. Therefore, we following Madry et al. (2018) and use 10-step Projected Gradient Descent to get the inner maximizer.

As can be seen from Figure 5, the empirical results on 2-layer ReLU network suggest very similar trends as the linear classifier for both adversarial risks and standard risks. This further backs up our theoretical conclusions.

