# OpenReview forum: "Benign Overfitting in Adversarially Robust Linear Classification"
_ICLR.cc/2022/Conference — ICLR 2022 Submitted_

### Official Review · Reviewer_V6MB · 2021-11-02

**Correctness:** 3
**Technical Novelty And Significance:** 3
**Empirical Novelty And Significance:** 3
**Recommendation:** 5
**Confidence:** 3

**Main Review:**

The problem discussed in this paper is very essential in machine learning, and the method of analysis will be useful for future reference.
The authors show that the upper bound of adversarial risk under adversarial training becomes similar to the standard risk under adversarial training.

However, I could not how the theoretical result relates to the "benign" or optimality.
- In this context, what does the "benign" means?
- Are there lower bounds on both of adversarial risk under adversarial training and the standard risk under adversarial training?
- Do the upper bound match the lower bounds? I would like to see a clearer explanation of the connection between the two.

The result just implies that the upper bound under adversarial risk matches that under standard risk, but this does not imply the "benign overfitting" because there is a possibility that the upper bounds are not tight.

Minor:
Holders' -> Hölder's?

**Summary Of The Paper:**

Learning with an over-parametrized model is an important problem in machine learning, and the benign overfitting problem garnered attention. The results shown in this paper has a significant implication in this context. In this paper, the authors investigate the phenomenon of benign overfitting in adversarial training. Surprisingly, the authors report that the benign overfitting phenomenon can be observed even in the case where we use adversarial training. The authors' approach is reasonable and should serve as a reference for future research.

**Summary Of The Review:**

This paper studies a very interesting setting. The approach is meaningful and can be used for future reference. However, I could not understand the extent to which the authors' claim of "benign overfitting" is supported. Therefore, I vote weak reject. If this point becomes clear to me and readers, I will raise the score to acceptance.

---

> ### Author Response · Authors · 2021-11-22
> **Response to Reviewer V6MB**
>
> Thank you for your constructive and helpful comments on our work!
>
> ---
> *Q1:* In this adversarial context, what does "benign" means?
> *A1:* In the adversarial context,  “benign” means that even an adversarially trained classifier memorizes the noisy training data, it can still achieve a good generalization performance on the adversarial examples.
>
> ---
> *Q2:* Are there lower bounds on both adversarial risk under adversarial training and the standard risk under adversarial training? Do the upper bound match the lower bounds?
> *A2:* For now, we are not aware of any lower bound for our setting. The difficulty for deriving such lower bounds comes from (i) the additional label flipping noises in our setting, and (ii) the complexity of adversarially robust classifiers. To our knowledge, even for logistic regression without adversarial training, there is no such lower bound in the presence of label flipping noises. Note that when taking $\epsilon \rightarrow 0$, $t\rightarrow \infty$, our result recovers the risk bound in Chatterji & Long (2020), which to a certain extent demonstrates the tightness of our results.
>
> In addition, we believe that a lower bound can be derived under a special setting without the label flipping noise (i.e., $\eta = 0$) and when we consider $\ell_2$ adversarial perturbations. In this setting, we can prove the equivalence between $\ell_2$ robust logistic regression and the $\ell_2$-robust maximum margin classification, and study the support vectors in the robust maximum margin classifier to derive an explicit form solution. Then an adversarial risk lower bound can be derived based on this explicit form solution. Similar techniques have been used in [*] to establish standard risk lower bound for maximum margin classifiers.
>
> In general, we believe deriving such a lower bound for $\ell_p$ perturbation is still an open problem and is an important future work direction.
>
> [*] Yuan Cao, Quanquan Gu, and Mikhail Belkin. Risk bounds for over-parameterized maximum mar-gin classification on sub-gaussian mixtures. NeurIPS2021 ​​
>
> ---
> *Q3:* The result just implies that the upper bound under adversarial risk matches that under standard risk, but this does not imply the "benign overfitting" because there is a possibility that the upper bounds are not tight.
> *A3:* We feel that this might be a misunderstanding of the reviewer. In order to illustrate benign overfitting, a diminishing risk upper bound suffices. This is very different from showing the “double-descent” phenomenon, where a precise risk curve (or a matching upper and lower bound) is required.  Recall that in our paper, we have shown in Remarks 4.6 and 4.10 that, with sufficiently large $d$ and under mild conditions, both the adversarial risk and the standard risk will converge to the optimal value $\eta$. This is sufficient to show that adversarial training can have “benign overfitting” properties.
>
> ---
> *Q4:* Minor: Holders' -> Hölder's?
> *A4:* Thanks for pointing this out. We have corrected it in the revision.

---

### Official Review · Reviewer_AeLK · 2021-11-02

**Correctness:** 4
**Technical Novelty And Significance:** 2
**Empirical Novelty And Significance:** 2
**Recommendation:** 6
**Confidence:** 4

**Main Review:**

Strength

The behaviour of overparameterised models in the presence of noise when trained with adversarial training is certainly a very important problem as both of these components(overparameterisation and noise) are important components in the ML context.

The analysis is clean and simple. The results are interpretable and  confirm the intuitions that one has about this problem,

Weaknesses

The proofs rely very heaviliy on Li. et. al. 2020 and Chhaterji et. al. 2020, which raises questions about the technical novelty of the paper. In particular it relies heavily on the characterisation of the GAT solution in Li. et. al. 2020 and the characterisation of test error in the presence of noise in Chatterji et. al. 2020.

The paper also deals with a simple linear model and it is not immediately clear how it relates to overparameterised neural networks. While it is important to first understand a complex phenomenon theoretically on a linear model, in the absence of  a huge amount of novelty in the theory, I find this second drawback rather glaring.


[1] "Implicit bias of gradient descent based adversarial training on separable data", Yan Li, Ethan X. Fang, Huan Xu, and Tuo Zhao, ICLR 2020.

[2] "Finite sample analysis of interpolating linear classifiers in the overparameterized regime", Niladri S Chatterji and Philip M Long.


========Update=============

I have read the author's response. It didn't change my view on the paper as the authors did not provide any new information. However, I would still recommend acceptance of the paper based on my initial evaluation of the paper.  The problem is important and the results while, might appear a bit derivative and for a simple setting, it might be necessary to first look at this simple setting to get a better progress.

**Summary Of The Paper:**

This paper studies the clean test and adversarial test error obtained using   _Gradient Descent Adversarial Training_ (GDAT). The data distribution is the gaussian mixture model and the hypothesis class is lilinear classifiers.

In Theorem 4.4, the authors show that the clean test error is worse than than the inherent noise rate. However, benign overfitting still exists though a larger $\epsilon$ for the adversarial training hurts clean error.

In Theorem 4.8, the authors upper bound the robust test error obtained by GDAT. The result shows that adversarial error is certainly worse off than clean test error. Perhaps somewhat interesting this also shows that the perturbation radius needs to decrease with increasing $d$.



**Summary Of The Review:**

In short, I found the problem interesting and the solution clean and understandable.

However, the technical novelty in the work is relatively low and it mainly involves combining existing works in a smart way. I would urge the authors to either extend the theoretical results eg. by looking into more complex hypothesis classes or to improve the experimental work.

---

> ### Author Response · Authors · 2021-11-22
> **Response to Reviewer AeLK**
>
> Thank you for your positive comments on our work!
>
> ---
> *Q1:* The proofs rely heavily on Li. et. al. 2020 and Chatterji et. al. 2020, which raises questions about the technical novelty of the paper.
> *A1:* While our proof heavily relies on Li. et. al. 2020 and Chatterji et. al. 2020, it is by no means a trivial combination of their results. In comparison with Chatterji et. al. 2020, our work gives the risk bound of the classifier given by adversarial training with $t$ iterations, while Chatterji et. al. 2020 only establishes a risk bound for the limiting solution of logistic regression (i.e., the maximum margin solution). Therefore the result in Chatterji et. al. 2020 can be seen as a special case of our result when $\epsilon = 0$, $t \rightarrow \infty$. Compared with Li et al. 2020, our work studies the generalization performance of gradient descent while Li et al. only study the optimization properties of gradient descent.
>
> ---
> *Q2:* The paper also deals with a simple linear model and it is not immediately clear how it relates to overparameterized neural networks.
> *A2:* We believe that the study of adversarial training for simple linear models is a fundamental problem for the understanding of benign overfitting in adversarial training. Without a full understanding of how and when benign overfitting can occur for the simplest possible neural network—the linear model—it seems unlikely that we will find satisfying explanations for why overfitted deep neural networks trained by adversarial training can generalize. As our work is the first result for benign overfitting of adversarial training, we think we have made significant progress on this problem.
> Furthermore,  we have already conducted adversarial training experiments on 2-layer neural networks with ReLU activation functions, which can be found in Appendix C.2. The experiments showed the same empirical trends and our conclusions still hold for 2-layer neural networks on both adversarial risks and standard risks.

---

### Official Review · Reviewer_7uYN · 2021-11-03

**Correctness:** 4
**Technical Novelty And Significance:** 3
**Empirical Novelty And Significance:** 1
**Recommendation:** 5
**Confidence:** 2

**Main Review:**

The main claim of the paper is that benign overfitting occurs in the adversarial case as well--that is, even though the training data are overfit perfectly, one can still attain reasonable generalization error. However, in practice it is well-documented that overfitting is indeed worse for adversarially-trained models, so I think an adequate analysis in this case would explain this phenomenon. I was not able to get such an explanation from the paper, although I may have missed it.

Aside from explaining empirical results, another potential contribution of theory papers is introducing new proof techniques. The paper does not include a discussion of what new technical contributions or proof techniques it contributes, so it was difficult to assess this aspect. While there is a proof outline in Section 5, it is not clear what is novel and what follows the previous proofs.

**Summary Of The Paper:**

The paper considers the analysis of benign overfitting in linear regression, first described in Bartlett et al. (2020), and extends it to the case of adversarial linear classification. Previously, Chatterji & Long (2020) had studied the non-adversarial classification case, so the main new result is an extension to the adversarial case.

**Summary Of The Review:**

I did not feel the paper made significant empirical or explanatory contributions, and was unable to assess the theoretical contributions. I thus overall would recommend against accepting the paper.

---

> ### Author Response · Authors · 2021-11-22
> **Response to Reviewer 7uYN**
>
> Thank you for your valuable comments on our work!
>
> ---
> *Q1:* In practice, overfitting is worse for adversarially-trained models. Not able to get an explanation on this.
> *A1:* Rice et al. (2020) showed that overfitting leads to worse empirical robustness on real image distributions. However, this does not refute our result. We believe that the main reason is due to the different data distributions we are considering. Our goal is not to explain the “robust overfitting” phenomenon observed in previous studies on image data, but to point out a striking fact that adversarial training can be benign for certain distributions. Specifically, our theoretical studies showed that for linear models with sub-Gaussian mixture data, the overfitting effect is indeed benign. Our empirical results also verified that this is indeed true with not only linear models but also neural networks. We believe data distribution plays a very important role in adversarial training, and benign overfitting can occur for certain data distributions but not all.
>
> ---
> *Q2:* Discussion of what new technical contributions or proof techniques it contributes.
> *A2:* Thanks for your suggestion. Our technical contributions lie in a margin analysis throughout the training of the linear model by gradient descent. This enables us to obtain risk bounds for finite iterations $t$ in Theorems 4.4 and 4.8. In comparison, existing works such as Chatterji & Long (2020) only gave risk bounds for the limiting classifier when $t \rightarrow \infty$. We will highlight our technical contributions in the revision.

---

> > ### Comment · Reviewer_7uYN · 2021-11-29
> > **Response**
> >
> > Thank you for your response. Regarding Q1, I understand that the empirical observations do not refute your results. My point is that, given the theory produces predictions that are inconsistent with the empirical data, this is a sign that the assumptions underlying the theoretical setting are not realistic.
> >
> > Regarding Q2, from the explanation given I still don't really get a sense of what new technical tools are being introduced. What obstacle do Chatterji and Long's results run into, and what needs to be changed to fix this?

---

> > > ### Author Response · Authors · 2021-11-30
> > > **Response to Additional Comments**
> > >
> > > Thank you for your reply and additional questions.
> > >
> > > **"Regarding Q1, I understand that the empirical observations do not refute your results. My point is that, given the theory produces predictions that are inconsistent with the empirical data, this is a sign that the assumptions underlying the theoretical setting are not realistic."**
> > >
> > > For Q1, we want to argue that understanding simple settings such as the mixture of Gaussian distributions is also important for a comprehensive understanding of adversarial training. It can help us rethink whether some previous claims that we take for granted are universally applicable or not.
> > >
> > > For example, many previous works studied whether adversarial examples are inevitable or not (whether a robust classifier can exist or not). [1][2] showed that for real image distribution it is possible to learn robust classifiers (high robustness upper bound). Yet [3][4][5] showed that simple distributions such as a uniform distribution over a unit sphere or Gaussian distribution can imply no robust model exists (inevitable adversarial examples). Although [3][4][5] studied the simple and “not realistic”  distributions, we still believe that these works provide us with many insights into adversarial training.
> > >
> > > Similarly, in our case, although previous studies have shown that for real image distributions, overfitting may not be benign for adversarial training, we should not take it for granted and believe this claim is true in all cases. In fact, we (are the first to) show that benign overfitting can occur for adversarial training on certain distributions. This new finding provides a different perspective on understanding adversarial training and we believe our contributions are also significant.
> > >
> > >
> > > [1] Fawzi, et al. Adversarial vulnerability for any classifier. In Advances in Neural Information Processing Systems (NeurIPS), 2018.
> > > [2] Zhang, Xiao, et al. Understanding the intrinsic robustness of image distributions using conditional generative models. International Conference on Artificial Intelligence and Statistics (AISTATS), 2020.
> > > [3] Shafahi, Ali, et al. "Are adversarial examples inevitable?." ICLR 2019.
> > > [4] Gilmer, et al. Adversarial spheres. arXiv preprint arXiv:1801.02774, 2018.
> > > [5] Tsipras, Dimitris, et al. "Robustness may be at odds with accuracy." ICLR 2019.
> > >
> > > **"Regarding Q2, from the explanation given I still don't really get a sense of what new technical tools are being introduced. What obstacle do Chatterji and Long's results run into, and what needs to be changed to fix this?"**
> > >
> > > Compared with Chatterji and Long, 2020, our work makes two significant contributions: (1) we study the benign overfitting phenomenon on adversarially trained classifiers while Chatterji et. al. 2020 only establishes a risk bound for standard training; (2) we give the risk bound of the classifier given by adversarial training after $t$ iterations, while Chatterji et. al. 2020 only establishes a risk bound for the limiting solution of standard training (i.e., the maximum margin solution).
> > >
> > > For (1), we need to extend the risk bound from the standard classifier to adversarially trained ones. Note that the biggest challenge here is that due to different loss objectives of adversarial training, the resulting gradient and the final solution are also different. Since the final solution is no longer the same, we need to derive new properties of the final solution as shown in Lemmas 5.3 and 5.4, which are essential for the derivation of Lemma 5.5.
> > >
> > > For (2), to give the risk bound of an adversarially trained classifier after $t$ iterations, we take into account the training dynamics of the adversarial training procedure. Specifically, Lemma 7 in Chatterji and Long, 2020 only considers the inner product of $\mu$ and the maximum margin solution. We extend Lemma 7 into our Lemma 5.5 by considering the inner product of $\mu$ and t-step adversarially trained model parameters. This enables a refined term that depends on the current iteration t which is originated from the convergence rate of the t-step adversarially trained classifier.
> > >
> > > We hope this answers your questions and we are happy to answer any other questions you have.

---

### Official Review · Reviewer_kp1N · 2021-11-11

**Correctness:** 4
**Technical Novelty And Significance:** 3
**Empirical Novelty And Significance:** 1
**Recommendation:** 5
**Confidence:** 4

**Main Review:**

This paper studies the "Benign Overfitting" phenomena in the setting of linear classification of sub-gaussian mixtures. Precise risk upper bounds are provided for adv-training of linear classifiers under exponential loss.

Overall, this paper is well-written and the technical results look correct to me (although I did not check the details very carefully). The setting of interest is an important first step towards understanding overparameterized models - it generalizes the prior works to the sub-Gaussian mixture setting.

However, there are a few weaknesses, summarized below.

(1) Important references missing: there are a few recent papers that studied very similar settings which the authors weren't aware of:
ICML 2020, Sharp Statistical Guarantees for Adversarially Robust Gaussian Classification, by Chen Dan, Yuting Wei, Pradeep Ravikumar
https://arxiv.org/abs/2010.11213: Precise Statistical Analysis of Classification Accuracies for Adversarial Training, by Adel Javanmard, Mahdi Soltanolkotabi
https://arxiv.org/abs/2010.13275: Asymptotic Behavior of Adversarial Training in Binary Classification, by Hossein Taheri, Ramtin Pedarsani, Christos Thrampoulidis
All of these papers studied a Gaussian setting (which is a special case of the sub-Gaussian setting studied in this work). The latter two papers also worked on overparameterized models under d/n = \Theta(1) regime, which is slightly different from this paper (their results are stronger in certain aspects and weaker in some other aspects).

I think a detailed comparison with these works is very necessary and may help readers understand the contribution and novelty better.

In particular, I have a few questions regarding the differences with these papers:
(1a) In Dan et al. 2020 (mentioned above), the authors proposed a notion called "adversarial signal-to-noise ratio", which measures the difficulty of adversarially robust classification in the Gaussian setting (a special case of this work). If my calculation was correct, for isotropic Gaussian mixture with l_inf perturbations,
$$AdvSNR = \|\mu\|_2 - \epsilon \|\mu\|_1 $$
Since the minimum error is roughly exp(-O(AdvSNR^2)), which roughly translates to
$$\exp(-C*(\|\mu\|_2^2 - 2 \epsilon \|\mu\|_1\|\mu\|_2))$$
in the risk bound. However, in this work (theorem 4.4 and 4.8), the risk bound has the form of
$$\exp(-C*(\|\mu\|_2^2 - 4 \epsilon \|\mu\|_1) - C'\|\mu\|_2 )$$
Can you explain this discrepancy? Is this a typo, or something more fundamental, like (I) the difference between Gaussian and sub-Gaussian or (II) the specific behavior of adv-trained classifiers that are different from information-theoretically optimal classifiers?

(1b) In Javanmard et al. 2020 and Taheri et al. 2020, the authors of both papers analyzed the setting of Gaussian mixtures in the d/n=O(1) setting. However, the analysis in this work (see theorem 4.4) requires d>= C*n^2 log(n), which is quite heavily overparameterized. To me, the proportional d/n=O(1) setting appears to be much more natural (although this may be my personal taste), and I think the analysis in this work isn't showing the complete story due to this requirement - it will be great if we can understand the behavior of adv training in the sub-quadratic overparameterization regime.

(2) There doesn't appear to be much clear take-away from the theoretical results. While this paper is written clearly, it is a little too dry, and the lacking of take-aways limits the impact on the whole field of adversarial robustness. Besides, some of the behaviors that appeared in this work contradict empirical evidence for deep neural networks.
For example, one of the main messages conveyed by this paper is that overfitting is benign in adversarial training. However, this is clearly not true for deep neural networks, as observed by Rice et al. 2020. This discrepancy takes the validity/usefulness of the setting studied into question.


**Summary Of The Paper:**

This paper studies the "Benign Overfitting" phenomena in the setting of linear classification of sub-gaussian mixtures. Precise risk upper bounds are provided for adv-training of linear classifiers under exponential loss.

**Summary Of The Review:**

I like the theoretical results. However, some more detailed comparisons with prior works, along with take-aways for practitioners, are necessary to make this paper better.

---

> ### Author Response · Authors · 2021-11-22
> **Response to Reviewer kp1N**
>
> Thank you for your valuable and insightful comments on our work!
>
> ---
> *Q1:* Missing references
> *A1:* Thank you for pointing out these papers. We will comment on these works in the following response. We have also revised the paper and added a comparison in Appendix A with these works.
>
> ---
> *Q2:*  Can you explain the risk bound discrepancy with Dan et al. 2020?
> *A2:* Thanks for this insightful question. Our setting and the setting in Dan et al. 2020 are different in the following aspects.
> 1. We study a Gaussian mixture model with additional label flipping noises, while Dan et al. 2020 does not consider such label flipping noise.
> 2. More importantly, we study an estimator found by gradient descent that overfits the training data, while Dan et al. 2020 study a specific plug-in estimator which does not overfit the training data.
>
> Due to the above differences, there is a discrepancy in the risk bounds derived in both papers.
>
> ---
> *Q3:*   In Javanmard et al. 2020 and Taheri et al. 2020, the authors of both papers analyzed the setting of Gaussian mixtures in the d/n=O(1) setting. It will be great if we can understand the behavior of adv training in the sub-quadratic overparameterization regime.
> *A3:*  To our knowledge, these are indeed two different settings. In the $d / n = O(1)$ proportional limit setting studied in Javanmard et al. 2020 and Taheri et al. 2020, the data Gram matrix and the sample covariance matrix can be studied based on random matrix theory/Gaussian comparison inequalities/convex Gaussian min-max theorem. In contrast, the $d > \tilde{O} (n^2 )$ setting we consider is the setting where the sample covariance matrix is singular but the n x n “data Gram matrix” concentrates around its expectation. We agree that the $d / n = O(1)$ setting is also very interesting, but we believe that the $d > \tilde{O} (n^2 )$ setting is also representative as many models such as neural networks are indeed heavily overparameterized ($d = poly (n)$).
>
> ---
> *Q4:* Take-away from the theoretical results. Besides, some of the behaviors that appeared in this work contradict empirical evidence for deep neural networks.
> *A4:* The main takeaway of our work is a striking fact that unlike those observed in previous studies (e.g., Rice et al. 2020), overfitting in adversarial training can be benign for certain distributions. Specifically, our theoretical studies showed that for linear models with sub-Gaussian mixture data, the overfitting effect is indeed benign. Our empirical results also verified that this is indeed true with not only linear models but also neural networks. We argue that this does not contradict existing observations as data distribution plays a very important role in adversarial training as suggested by many previous studies [1][2][3] that different distributions can imply no robust model exists (inevitable adversarial examples [3]) or the other way around ([1][2]). It is possible that benign overfitting occurs in one distribution but not another.
>
> [1] Zhang, Xiao, et al. Understanding the intrinsic robustness of image distributions using conditional generative models. International Conference on Artificial Intelligence and Statistics (AISTATS), 2020.
> [2] Fawzi, et al. Adversarial vulnerability for any classifier. In Advances in Neural Information Processing Systems (NeurIPS), 2018.
> [3] Gilmer, et al. Adversarial spheres. arXiv preprint arXiv:1801.02774, 2018.

---

### Decision · Program_Chairs · 2022-01-20

**Decision:**

Reject

**Comment:**

The paper studies the benign overfitting phenomenon for linear models with adversarial training. The main issue is that the result is quite expected for experts versed in the benign overfitting papers, and indeed the reviewers pointed out that they could not see much technical novelty. However, even more importantly, the original benign overfitting papers had the advantage of proposing of simpler model (linear!!!) with the same behavior as the complex ones in practice. This is not the case here, as the result diverge from empirical observations on deep networks. The authors argue that it is a valuable finding that the empirical observation is not "universal", but this is a somewhat moot point as linear models are a priori very very different from the setting in which these empirical observations were made. For these reasons I believe the paper does not meet the bar for ICLR (yet it could still be publishable elsewhere).